# SAFE REINFORCEMENT LEARNING IN BLACK-BOX ENVIRONMENTS VIA ADAPTIVE SHIELDING

## ABSTRACT

Empowering safe exploration of reinforcement learning (RL) agents during training is a critical impediment towards deploying RL agents in many real-world scenarios. Training RL agents in unknown, *black-box* environments poses an even greater safety risk when prior knowledge of the domain/task is unavailable. We introduce ADVICE (Adaptive Shielding with a Contrastive Autoencoder), a novel post-shielding technique that distinguishes safe and unsafe features of state-action pairs during training, thus protecting the RL agent from executing actions that yield potentially hazardous outcomes. Our comprehensive experimental evaluation against state-of-the-art safe RL exploration techniques demonstrates how ADVICE can significantly reduce safety violations during training while maintaining a competitive outcome reward.

## 1 INTRODUCTION

Reinforcement Learning (RL) (Sutton and Barto, 2018) is a powerful machine learning paradigm for solving complex decision-making tasks that has exhibited performance commensurate with the cognitive abilities of humans in diverse applications, including game-playing (Silver et al., 2017; 2018; Berner et al., 2019) and robot control (Rudin et al., 2022; Heess et al., 2017). Despite this huge potential, developing RL-based agents that can explore their environment safely remains a significant challenge. Exploring unfamiliar, and potentially hazardous, states while learning from the environment, especially in safety-critical domains, like robotics or healthcare, can pose real dangers. Alleviating this entails RL agents capable of synthesising an optimal policy by exploring the policy space adequately while ensuring safe exploration by preventing the execution of unsafe actions (Amodei et al., 2016).

Ensuring safety becomes an increasingly difficult challenge in complex environments typically characterised by high-dimensional state/action spaces (Dalal et al., 2018). Such environments require a large amount of training time before the agent can consistently complete the task and avoid safety concerns. This issue is further exacerbated in *black-box* environments where no prior knowledge can be utilised before training; the only information available is the data observed in real time by the RL agent. In such scenarios, the risk associated with exploration increases exponentially as the agent must operate without pre-defined guidelines, rendering typical safe exploration techniques inadequate (Waga et al., 2022).

Prior research on safe RL exploration formulates safety constraints as linear temporal (Alshiekh et al., 2018; Könighofer et al., 2023; ElSayed-Aly et al., 2021) and probabilistic logic (Yang et al., 2023) specifications, whose use as a shield protects the agent during training. Shielding techniques are categorised into pre-shielding (restricting action choices to a predefined safe subset) and post-shielding (evaluating and modifying actions post-selection to ensure safety) (Odriozola-Olalde et al., 2023). Despite noteworthy advances, a common challenge across these methods is their reliance on some degree of prior knowledge about the environment, task, or safety concern which may not be generally available. Research targeting safe exploration in black-box environments employs Lagrangian methods (Stooke et al., 2020; Altman, 1998; Tessler et al., 2018; Achiam et al., 2017), or involves a pre-training phase before the shield synthesis (Tappler et al., 2022).

We present ADVICE (**AD**apti**V**e Sh**I**elding with a **C**ontrastive Auto**E**ncoder), a novel post-shielding technique for the safe exploration of RL agents in *black-box* environments. The ADVICE shield is underpinned by a contrastive autoencoder that effectively learns distinguishing latent representations

between safe and unsafe features (state-action pairs) and a non-parametric classifier that employs these latent representations, allowing new features to be identified as safe, or correcting them with a new safe action when deemed unsafe. Further, ADVICE encompasses an adaptation component that considers the agent's recent performance to automatically regulate the risk tolerance levels of the shield, thus encouraging exploration when appropriate. We demonstrate ADVICE's ability to work in highly complex *black-box* environments and significantly reduce safety violations during training in comparison to state-of-the-art methods, such as Lagrangian multipliers (Lillicrap et al., 2015), discretized shields (Shperberg et al., 2022) and conservative safety critics (Bharadhwaj et al., 2020). To the best of our knowledge, ADVICE is the first research work that investigates shielding for safe RL exploration in black-box environments with high-dimensional state/action spaces and introduces an end-to-end approach for shield synthesis without using *any* prior knowledge.

## 2 RELATED WORK

**Shielding Techniques.** Recent research devises techniques for the safe exploration of RL agents using safety shields, allowing the agent to select from a pool of safe actions or correct an action deemed unsafe (Odriozola-Olalde et al., 2023). Existing shielding approaches leverage linear temporal logic (LTL) specifications (Alshiekh et al., 2018; Könighofer et al., 2023; ElSayed-Aly et al., 2021) or use external hints for constructing the LTL formulae (Waga et al., 2022). LTL specifications can be replaced with probabilistic logic programming (PLP) (Yang et al., 2023), extending their use to continuous deep RL and enabling safety constraints to be differentiable. By utilising logical neural networks (Kimura et al., 2020), the same logic specifications can be both respected and learnt, providing a more nuanced understanding of safety. Jansen et al. (2020) introduce probabilistic shields to ensure safety, while other approaches implement safety under partial observability (Carr et al., 2023) or use approximate models of the environment to maintain safety (Goodall and Belardinelli, 2023). A recurring limitation of these methods is their reliance on explicit prior knowledge of their environment, task, and/or safety concerns. Although they can improve safety and, in some cases, eliminate violations altogether, their applicability is restricted to a narrow set of environments and safety considerations (Turchetta et al., 2020). In contrast, our ADVICE post-shielding method does not need *any* prior knowledge, using exclusively the information captured in a typical RL problem.

**Black-Box Safe Exploration Techniques.** Other recent research focuses on improving safety in *black-box* environments, where no prior knowledge is provided to the agent/user. A trivial but effective solution is to record all unsafe features in a tabular format to prevent the agent from repeating them (Shperberg et al., 2022). However, this approach is limited to discrete environments or extremely low-dimensional spaces. Other research collects data in the environment before training, to then utilise a safety layer (Dalal et al., 2018; Srinivasan et al., 2020; Thananjeyan et al., 2021; Bharadhwaj et al., 2020) or shield (Tappler et al., 2022) to protect the agent. This requires a significant amount of data collection, resulting in an increasing number of safety violations. Lagrangian methods enable modelling the Markov Decision Process as a Constrained Markov Decision Process, leading to their wide adoption due to their simplicity and effectiveness (Altman, 2021; Garcıa and Fernández, 2015). The Lagrangian multiplier can be fixed (Stooke et al., 2020; Altman, 1998), or integrated into the algorithm itself (Tessler et al., 2018; Achiam et al., 2017). Other advances employ uncertainty estimation concepts (Kahn et al., 2017; Jain et al., 2021). Defining safety in terms of uncertainty and propagating it into the RL algorithm during training yields a cautious yet effective agent for reducing safety violations, even in complex environments. Similarly to these approaches, ADVICE needs no prior knowledge; however, ADVICE also entails far less data collection than the previously discussed techniques.

## 3 PRELIMINARIES

**Markov Decision Process.** A Markov Decision Process (MDP) (Bellman, 1957) is a discrete-time stochastic control process to model decision-making. An MDP is formally defined as a 5-tuple $M = (S, A, P, R, \gamma)$, where $S$ is the state space, $A$ is the action space, $P$ is the state transition probability matrix such that $P(s'|s, a)$ is the probability of transitioning to state $s'$ from state $s$ using action $a$, $R$ is the reward function such that $R(s, a)$ is the reward for taking action $a$ in state $s$, and $\gamma$ is the discount factor that determines the value of future rewards. A policy $\pi : S \to \Delta(A)$ is a distribution over actions given a state. MDPs can be solved using dynamic programming

techniques (e.g., value iteration, policy iteration) which require complete knowledge of the MDP's dynamics (Bertsekas and Tsitsiklis, 2008).

**Reinforcement Learning.** Reinforcement Learning (RL) involves training an agent to make a sequence of decisions by interacting with the MDP, which represents the environment. This machine learning technique is used to solve MDPs when full knowledge is not available. The agent's goal is to find a policy $\pi^*$ maximising the expected discounted return $E\left[\sum_{t=0}^{\infty} \gamma^t R_{a_t}(s_t, s_{t+1})\right]$ (Sutton and Barto, 2018). The value function $V_\pi(s)$ informs the agent how *valuable* a given state is when following the current policy $\pi$. Common RL algorithms include Q-learning (Watkins and Dayan, 1992), and SARSA (Rummery and Niranjan, 1994). Deep reinforcement learning (DRL) extends traditional algorithms by utilising deep neural networks to approximate the policy $\pi$ or the value function $V$ when the state/action space is high-dimensional and complex. Actor-critic methods (Sutton and Barto, 2018) are a popular class of algorithms both in traditional and deep RL. Distinctly, the policy (actor) and the value function (critic) are modelled as separate components, allowing for simultaneous updates to both functions. Deep Deterministic Policy Gradient (DDPG) (Lillicrap et al., 2015) is an example of an actor-critic method tailored specifically for continuous action spaces.

**Contrastive Learning.** Contrastive Learning (CL) (Hadsell et al., 2006) is an unsupervised or semi-supervised machine learning paradigm aiming at distinguishing between similar (positive) and dissimilar (negative) pairs of data. At its core lies a contrastive loss function, which encourages the model to put similar pairs closer together in the embedding space while separating dissimilar pairs. Given a pair of inputs $x_i$, $x_j$, the contrastive loss function is defined as:

$$L(x_i, x_j, y, \theta) = y \cdot \|h_\theta(x_i) - h_\theta(x_j)\|^2 + (1 - y) \cdot max(0, m - \|h_\theta(x_i) - h_\theta(x_j)\|^2) \quad (1)$$

where the binary label $y$ indicates if the pair is similar ($y = 1$) or dissimilar ($y = 0$), $h$ is the embedding vector and the margin $m$ regulates the minimum distance between dissimilar pairs. The loss function encourages the model to learn meaningful representations that reflect the inherent similarities and differences between data points, thus facilitating the formation of well-defined clusters in the embedding space.

## 4   ADVICE

Our ADVICE post-shielding technique empowers safe RL exploration in black-box environments without requiring a system model. Figure 1 shows a high-level overview of ADVICE, including its key shield construction, execution and adaptation stages, and its incorporation within the standard RL loop. The core of ADVICE comprises a contrastive autoencoder (CA) model that can efficiently distinguish between safe and unsafe features from the feature space $\mathcal{F} : S \times A$ (representing all possible state-action pairs). A feature $f_t = (s_t, a_t)$, $f_t \in \mathcal{F}$, denotes a state-action pair at timestep $t$. The CA model leverages a unique loss function where similar and dissimilar features are compared, enabling the systematic identification of meaningful latent feature representations. ADVICE employs these latent representations and specialises an unsupervised nearest neighbours model to the learnt embedding space, thus enabling the classification of new features. Formally, the ADVICE shield is defined by the function

$$\phi : F \times \mathbb{Z}_+ \to A \quad (2)$$

such that at time step $t$ during the execution of an episode, ADVICE evaluates the agent's desired action $\phi(f_t, K)$ and allows the action $a_t$ to be taken, or enforces the execution of another safe action $a'_t$ instead. The ADVICE-specific parameter $K$ enables controlling the risk aversion levels of the shield. In particular, ADVICE considers the agent's performance and automatically adapts the $K$ value, thus supporting the dynamic calibration of ADVICE's cautiousness level during learning. Next, we introduce the details of ADVICE and its integration within the RL loop as a post-shield.

### 4.1   ADVICE SHIELD CONSTRUCTION

As a black-box post-shielding technique, ADVICE does not rely on any prior knowledge about the RL agent or its environment. Instead, the ADVICE shield construction stage is founded on collecting a feature set $F_E$ during an initial unshielded interaction period of $E$ episodes where the RL agent is allowed to interact with and collect experience from its environment. A feature $f_t \in F_E$ is classified

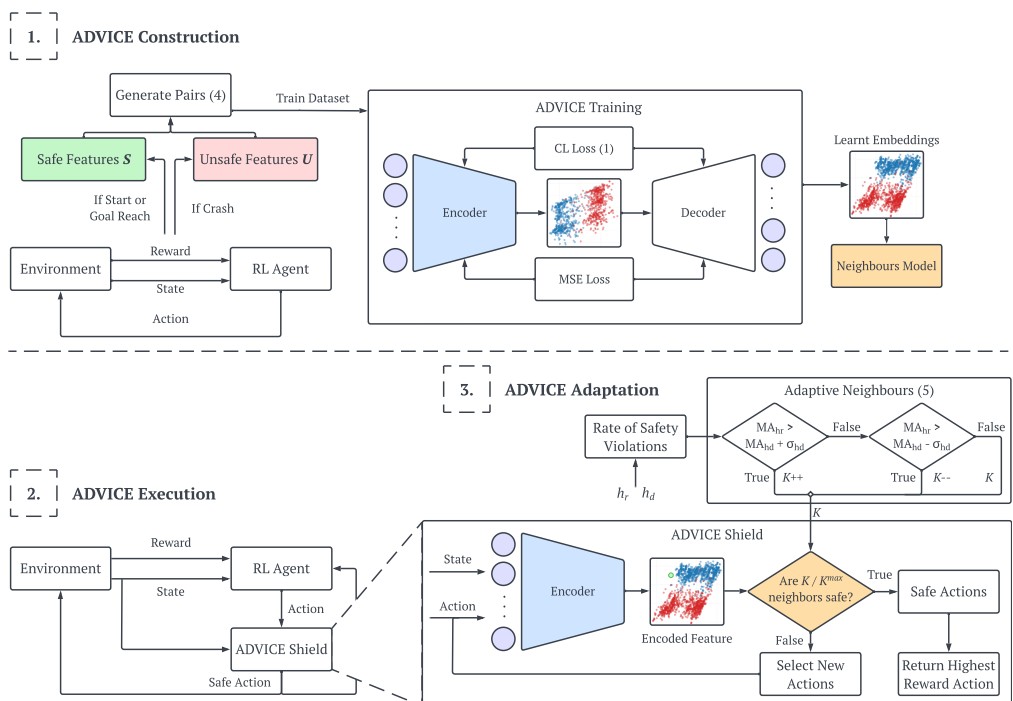

Figure 1: A high-level overview of the ADVICE construction, execution, and adaptation.

as either safe $\mathcal{S} \subset F_E$, unsafe $\mathcal{U} \subset F_E$ or inconclusive $\mathcal{I} \subset F_E$ based on the following function:

$$g(f_t) = \begin{cases} \text{safe} & \text{if } t = 0 \text{ or } s_{t+1} = \text{goal (accepting state)} \\ \text{unsafe} & \text{if } s_{t+1} = \text{terminal (failure state)} \\ \text{inconclusive} & \text{otherwise} \end{cases} \quad (3)$$

where $S = \{f_t \in F_E | g(f_t) = \text{safe}\}$, $U = \{f_t \in F_E | g(f_t) = \text{unsafe}\}$, and $I = F_E \backslash (S \cup U)$. Hence $\mathcal{S} \cup \mathcal{U} \cup \mathcal{I} = F_E$ and $\mathcal{S} \cap \mathcal{U} \cap \mathcal{I} = \varnothing$. The collected features set $F_E$ from the initial interaction period $E$ are organised into these categories to facilitate the contrastive learning process: (i) pairs of features that are similar (e.g., two safe/unsafe features); and (ii) feature pairs that are dissimilar (a safe and an unsafe feature). This categorisation is vital to allow the model to discern between safe and unsafe features effectively and focus on finding meaningful representations in a lower-dimensional latent space that reflect the similarities and differences. The combine function $\mathcal{C}$ consumes the set of features in sets $\mathcal{S}$ and $\mathcal{U}$ and produces all pairwise combinations from $\mathcal{S}$ and $\mathcal{U}$:

$$\begin{aligned} \mathcal{C}(\mathcal{S} \cup \mathcal{U}, \mathcal{S} \cup \mathcal{U}) = &\{(g(f_t), g(f_{t'}), 1) \mid g(f_t) = g(f_{t'}) = \text{safe}, f_t \neq f_{t'}\} \cup \\ &\{(g(f_t), g(f_{t'}), 1) \mid g(f_t) = g(f_{t'}) = \text{unsafe}, f_t \neq f_{t'}\} \cup \\ &\{(g(f_t), g(f_{t'}), 0) \mid g(f_t) = \text{safe}, g(f_{t'}) = \text{unsafe}\} \end{aligned} \quad (4)$$

where $(g(f_t), g(f_{t'}), 1)$ is a pair of similar features, and $(g(f_t), g(f_{t'}), 0)$ shows dissimilar features.

The training of the CA model involves using the pairs of collected features and optimising two loss functions simultaneously: the mean squared error (MSE) and the contrastive loss function (CL) presented in Equation (1). The MSE loss measures how accurately the model can reconstruct the input features after encoding and decoding. In contrast, the CL is designed to refine the model's ability to cluster similar features together, whilst separating dissimilar ones within the latent space based on the Euclidean distance between them. This distance is minimised for similar pairs and maximised for dissimilar pairs, encouraging high cohesion and high separation between similar and dissimilar features, respectively. Thus, the CA learns to encode and reconstruct the salient features of the given RL problem, comprising the state of the environment and chosen action, as accurately as possible in a lower-dimensional latent space.

Once trained, the CA model is adept at finding nuanced distinctions between safe and unsafe features and accurately placing unseen features within the appropriate partitions in the latent space. The shield

construction stage of ADVICE concludes with embedding an unsupervised nearest neighbours (KNN) model in the latent space that classifies new encoded features as safe/unsafe. A visual representation of the latent data encoding, illustrating the clear separation in the latent space, is shown in Figure 2.

## 4.2 ADVICE EXECUTION AND ADAPTATION

Upon the completion of the shield construction phase, ADVICE can be used as a post-shield within the RL loop. The constructed shield encourages safe environment exploration, guarding the agent throughout its interaction with the environment by ensuring the execution of safe actions.

The ADVICE execution and adaptation stage, shown in Algorithm 1 and Figure 1 (bottom), involves a continuous cycle of action evaluation and decision-making until the maximum number of training episodes $E_{max}$ is reached (line 1). During an episode's execution (line 3), the RL agent, having observed the current state $s_t$, selects an action $a_t$ based on its current policy $\pi$ (line 4). Without ADVICE, the agent would proceed with this action regardless of its potential safety implications. Instead, ADVICE can now be used to evaluate the selected action before it is realised by the agent, thus ensuring safe environment exploration. More specifically, to establish if the action $a_t$ is safe, ADVICE extracts the latent representation $\hat{f}$ of feature $f_t$, collects the nearest $K^{max}$ latent data points to $\hat{f}$ and checks whether the cardinality of the safe data points exceeds the safety threshold $K$, in which case the action $a_t$ is considered safe; otherwise, $a_t$ is considered unsafe (lines 13–17).

The safety threshold $K \in [0, K^{max}]$ determines how many neighbours need to be labelled safe so that the encoded feature can be deemed safe. The closer $K$ is to $K^{max}$ entails that ADVICE will be more cautious in terms of safety, while a $K$ value closer to $\lceil K/2 \rceil$ means that the ADVICE shield is more relaxed and favours exploration. A value of $K < \lceil K/2 \rceil$ should not be considered as this would allow the RL agent to execute actions that are more likely to be unsafe than safe.

If action $a_t$ is deemed unsafe (line 5), ADVICE intervenes and generates a set of valid candidate actions $A_{f_t}$ by quantising the continuous space of each action dimension $A^d$, where $d \in D$ is the dimension, and extracting the Cartesian product across the $D$ action dimensions (line 6). The set of candidate actions $A_{f_t}$ is then filtered, to retain only valid and safe actions, resulting in $A'_{f_t} \subseteq A_{f_t}$ (line 7). If the $A'_{f_t}$ set is not empty, these filtered candidate actions are evaluated by the RL agent's value function $Q_\pi$ for their expected reward (line 9). The action with the highest expected reward is selected, entailing that the action aligns with both the safety considerations and the agent's

---

**Algorithm 1** ADVICE Execution and Adaptation

**Require:** Contrastive Autoencoder $CA$, Neighbours $K^{max}$, Recent history $h_r$, Distant history $h_d$
1: **while** $E{+}{+} \leqslant E_{max}$ **do**
2:      $s_1 \leftarrow$ OBSERVE()
3:      **for** $t = 1, \ldots, T$ **do**
4:          $a_t \leftarrow \pi(s_t)$
5:          **if** !ISSAFE($f_t, K, K^{max}$) **then**
6:              $A_{f_t} \leftarrow \prod_{d \in D} A^d_{f_t} | A^d_{f_t} \subset A^d$
7:              $A'_{f_t} \leftarrow \{a | \forall a \in A_{f_t} : f'_t = (s_t, a) \bullet$ ISSAFE($f'_t, K, K^{max}$)$\}$
8:              **if** $A'_{f_t} \neq \varnothing$ **then**
9:                  $a_t \leftarrow \operatorname{argmax}_{a \in A'_{f_t}} Q_\pi(s_t, a)$
10:         $r_t, s_{t+1} \leftarrow$ EXECUTE($a_t$)
11:         $\pi \leftarrow$ UPDATE_POLICY($a_t, s_t, r_t, s_{t+1}$)
12:         $K \leftarrow$ UPDATE_CAUTIOUSNESS($K, K^{max}, h_d, h_r$)             $\rhd$ (5)-(7)

13: **function** ISSAFE($f, K, K^{max}$)
14:      $\hat{f} \leftarrow$ ENCODE($f$)             $\rhd$ Encode feature f into the latent space
15:      $N_{\hat{f}} \leftarrow$ GETNEIGHBOURS($\hat{f}, K^{max}$)
16:      **if** $\sum_{n \in N_{\hat{f}}} [n == "safe"] \geqslant K$ **return** True          $\rhd$ Action considered safe
17:      **return** False             $\rhd$ Action considered unsafe

performance objectives. Note that if no safe alternative action is identified (i.e., $A'_{f_t} = \varnothing$), ADVICE resorts to providing the originally selected action $a_t$; since no alternative action is predicted to be safe, $a_t$ is anticipated to achieve the highest expected reward.

At the end of each episode, ADVICE calibrates its cautiousness level by considering the recent safety performance of the RL agent (line 12). This unique ADVICE characteristic enables moving beyond the static definition of the safety threshold $K$ by automatically adjusting its value in response to the frequency of recent safety violations. Accordingly, ADVICE becomes adaptive and allows the agent to explore more when exhibiting safe behaviour for a period of time while being more cautious, thus interfering more, when the RL agent behaves increasingly unsafely.

To assess the agent's performance over time, ADVICE employs a double sliding window, commonly used in the field of anomaly detection (Tu et al., 2019; Wang et al., 2024). By comparing the recent rate of safety violations against a broader historical view, ADVICE can discern whether the current trend deviates from the pattern seen historically. This analysis informs ADVICE whether to strengthen or relax the cautiousness level. The adaptation function is defined as follows:

$$K = \begin{cases} min(K+1, K^{max}), & \text{if } MA_{h_r} > MA_{h_d} + \sigma_{h_d} \\ max(K-1, \lceil K^{max}/2 \rceil), & \text{if } MA_{h_r} < MA_{h_d} - \sigma_{h_d} \\ K, & \text{otherwise} \end{cases} \quad (5)$$

where $MA_{h_d}$ and $\sigma_{h_d}$ are the moving average and standard deviation, respectively, of safety violations $\mathcal{Z}_i$ based on the distant history window $h_d$, and $MA_{h_r}$ is the moving average of $\mathcal{Z}_i$ using the recent history window $h_r$, i.e., $h_r < h_d$. Given $h \in \{h_r, h_d\}$, the calculation of the moving average $MA_h$ and standard deviation $\sigma_h$, considering the history of safety violations $\mathcal{Z}_1, \ldots, \mathcal{Z}_t$, is defined by:

$$MA_h = \frac{1}{h} \sum_{i=t-h}^{t} (\mathcal{Z}_i - \mathcal{Z}_{i-1}) \quad (6) \qquad \sigma_h = \sqrt{\frac{1}{h} \sum_{i=t-h}^{t} ((\mathcal{Z}_i - \mathcal{Z}_{i-1}) - MA_h)^2} \quad (7)$$

If the $MA_{h_r}$ rate exceeds the sum of $MA_{h_d} + \sigma_{h_d}$, then it signifies that the agent crashes more often than expected. Consequently, $K$ is automatically incremented to adopt a more cautious stance. Inversely, if the $MA_{h_r}$ rate is below $MA_{h_d} - \sigma_{h_d}$, this suggests that the agent acts more conservatively than expected. As a result, $K$ is automatically reduced to allow the agent to explore more freely. The safety threshold $K$ remains the same for any other occasion. ADVICE deliberately only considers one standard deviation as two or more standard deviations would make the adaptation slow to respond to emerging safety risks. Our sensitivity analysis on these two parameters (Section 5.4), demonstrates the described behaviours.

## 5 EVALUATION

We evaluate ADVICE [1] against state-of-the-art safe exploration methods using tasks from the Safety Gymnasium test-suite (Ji et al., 2023a), where a robot with Lidar sensors must complete tasks by navigating through obstacle-filled environments. This benchmark is designed to assess performance and safety in complex, high-dimensional scenarios with various safety constraints. Accordingly, the Safety Gymnasium is an ideal benchmark to assess the effectiveness of ADVICE and has been used by comparable techniques (Bharadhwaj et al., 2020). A terminal state is reached when the agent collides with an obstacle, signifying catastrophic damage to the robot and ending the episode. The selected tasks below cover a range of complexities in agent, goal, and obstacle positions:

- **Semi-random Goal**: A standard goal environment with six obstacles and one goal. The obstacles have a static spawn, while the agent and the goal have randomised positions every episode.
- **Randomised Goal**: A similar goal environment, but obstacles have random positions per episode.
- **Randomised Circle**: The agent has to circle in a given zone in this environment. The aim is to maximise speed and distance from the zone's centre while avoiding the three randomised obstacles.
- **Constrained Randomised Goal**: A randomised goal variant environment where the obstacles are hazards that impose a step-wise cost when the agent is inside. The objective is to minimise the cost.

Further details of these four environments and the corresponding tasks are provided in Appendix A. Figure 7 in this appendix shows a snapshot of the tasks within the Safety Gymnasium. To evaluate the performance of ADVICE, we conduct comparisons against the following state-of-the-art algorithms:

[1]The ADVICE code is available at: https://anonymous.4open.science/r/ADVICE-6AF9

- **DDPG**: A deep deterministic policy gradient (DDPG) agent (Lillicrap et al., 2015) which is the foundational baseline.
- **DDPG-Lag**: A DDPG agent with an online Lagrangian multiplier ($\lambda = 0.1$, $\alpha = 0.01$) which heavily penalises constant safety violations (Borkar, 2005) dynamically.
- **Tabular Shield**: A DDPG agent with a discretised (1 decimal place) table of terminal state-action pairs, which prevents the agent from executing inadequate actions again (Shperberg et al., 2022).
- **Conservative Safety Critic (CSC)**: A DDPG agent with a conservative safety critic ($\epsilon = 0.3$, $\beta = 0.2$) that uses conservative estimates to evaluate the safety of actions (Bharadhwaj et al., 2020).

Each method employs the same DDPG configuration, ensuring a fair and accurate comparison. Detailed information per algorithm structure and hyperparameter settings for all experiments is available in Appendix F. The results are averaged across three independent runs and include mean scores and confidence intervals (standard error of the mean), providing a performance overview. Unless explicitly stated, the default ADVICE deployment per run is $E = 1000$ with $K = 4$ and $K^{max} = 5$. To objectively evaluate all safe RL methods mentioned above, the setup setting employed is common for all, i.e., unconstrained, black-box MDPs. However, this setting can lead to sparse data due to the lack of frequent constraint violations. Therefore, to demonstrate the applicability of ADVICE in Constrained MDP environments (Altman, 2021) tailored for the DDPG-Lag and CSC methods, we also assess in Section 5.3 all methods in a constrained environment (i.e., Constrained Randomised Goal). Since this constrained environment aligns with the original design of DDPG-Lag and CSC, it enables the examination of ADVICE's performance in the setting particularly suited for DDPG-Lag and CSC. Theoretical analysis for ADVICE can be found in Appendix B.

## 5.1 PERFORMANCE RESULTS

For our first set of experiments, we evaluate the overall performance of the examined methods (DDPG, DDPG-Lag, Tabular shield, CSC, ADVICE) for the semi-random goal, randomised goal and randomised circle environments. Figure 2 shows the average episodic reward (return), cumulative safety violations (failed episodes), and cumulative goal reaches (successful episodes), alongside an example latent space visualisation from a single ADVICE execution. The obtained results provide evidence that ADVICE, albeit designed to prioritise safety, manages to maintain a competitive performance (reward) in all tasks. Both DDPG-Lag and ADVICE, indicate an inherent trade-off between reward maximisation and safety prioritisation.

Since the primary goal of a safety shield is to encourage exploration and learning in a safe way, it is evident that ADVICE, across all tasks, can significantly reduce safety violations, outperforming by a notable margin all other methods. DDPG-Lag also manages to reduce safety violations, but not to the magnitude of ADVICE, highlighting the effectiveness of ADVICE's shield. Through all three tasks, it is noticeable that the conservative safety critic (CSC) vastly underestimates safety. This is due to the sparse data that comes with an unconstrained MDP mixed with the complex safety constraints. Without a consistent cost signal, regular neural network-based safety techniques struggle to keep the agent safe. This insight further highlights the strength of the contrastive learning underpinning ADVICE. We visualise the learnt latent space from an example ADVICE run (Figure 2 right), illustrating the power of contrastive learning and validating its effectiveness at learning latent representations that empower the distinction between safe and unsafe features. Considering that ADVICE constructs the shield for the first $E = 1000$ episodes and then starts its execution for safe exploration, while DDPG-Lag aims at reducing safety violations from episode $E = 1$, the results are even more affirmative.

Despite its conservative prioritisation to reward, ADVICE still reaches the goal (successful episode execution) with a similar frequency to all other methods. In fact, ADVICE completes the task without compromising safety, and the difference in accumulated reward is due to the reward function design and the reduced time the agent has to complete the task using a cautious path. Through additional experiments with an increased number of maximum steps per episode ($E' = 2000$ – Figure 14 - Appendix C), we have established that ADVICE's average episodic return is very similar to the other methods. Hence, increasing the episode's duration yields improved ADVICE-based results.

These core results corroborate that ADVICE's contrastive approach to shielding an RL agent from unsafe actions is effective in comparison to other methods. ADVICE significantly reduces safety violations without detracting much from the agent's overall performance.

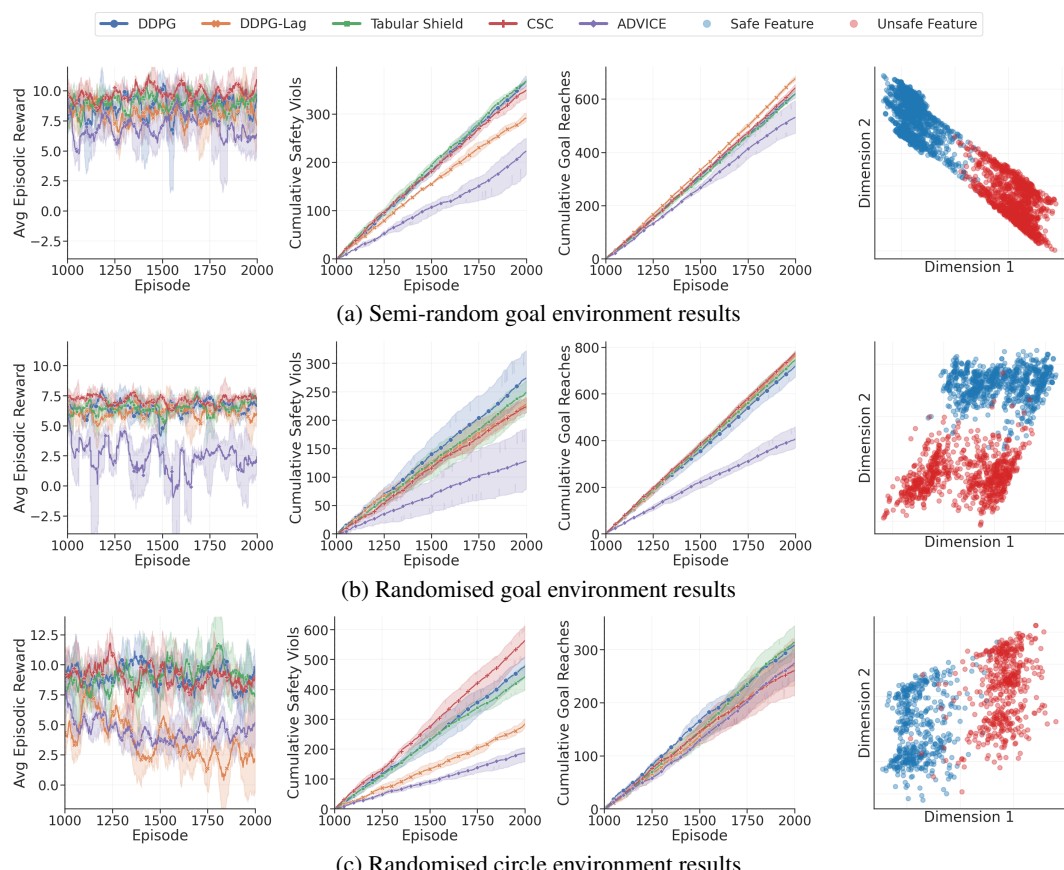

Figure 2: Average episodic reward, cumulative safety violations, cumulative goal reaches of examined methods (DDPG, DDPG-Lag, Tabular shield, Conservative Safety Critic, ADVICE) and example latent space visualisation for the semi-random goal (top), randomised goal (middle) and randomised circle (bottom) environments.

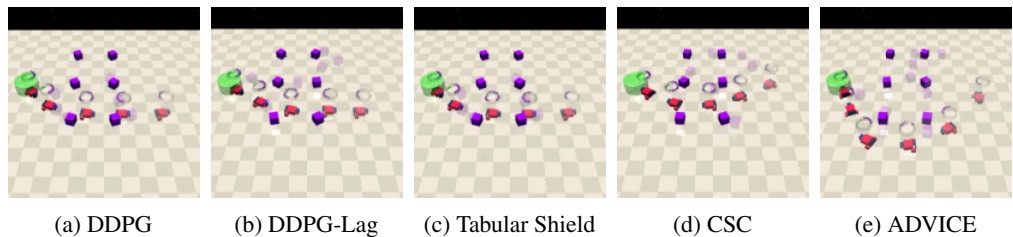

(a) DDPG      (b) DDPG-Lag      (c) Tabular Shield      (d) CSC      (e) ADVICE

Figure 3: Example trajectories of the DDPG, DDPG-Lag, Tabular Shield, Conservative Safety Critic, and ADVICE RL agents on the semi-random goal environment; obstacles are purple, goals are green circles, and the RL agent is the red vehicle.

Figure 3 visualises example trajectories of the evaluated methods on the semi-random goal environment, further validating our core findings. The DDPG and Tabular shield agents focus primarily on maximising expected return, and thus, navigate through the centre of the wall of objects to reach the goal. Alongside action noise, this is risky as the agent may collide with an object, as shown in Figure 2a. The DDPG-Lag and CSC agents tend to learn similar behaviours, except they leave a larger gap between themselves and the objects, considering the possibility of action noise. Finally, ADVICE adopts a cautious approach, learning to navigate the long way around the wall of objects. Learning this path ensures that the agent is less likely to collide with an obstacle. However, this incurs a longer trajectory and a slightly reduced reward due to moving further away from the goal initially.

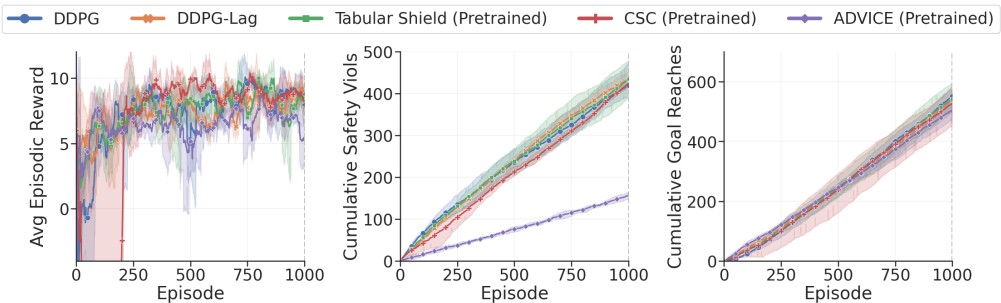

Figure 4: Average episodic reward, cumulative safety violations, and cumulative goal reaches on the semi-randomised goal environment where ADVICE , Tabular Shield, and CSC have a pre-trained shield.

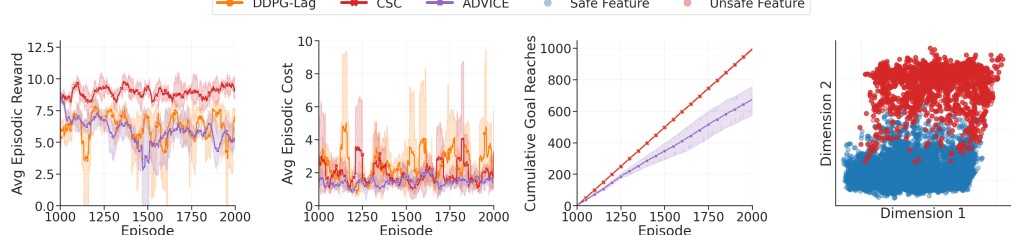

Figure 5: Average episodic reward, average episodic cost, cumulative goal reaches, and example latent space visualisation on the constrained randomised goal environment.

## 5.2 TRANSFER LEARNING

Figure 4 demonstrates the transfer learning capabilities of ADVICE in a semi-random environment. ADVICE, along with Tabular Shield and CSC, utilizes a pre-trained shield from a randomized environment and is then deployed in a new setting with different goal and agent positions. Leveraging this pre-trained shield, ADVICE significantly reduces safety violations from the outset, unlike the other methods. Over 1000 episodes, ADVICE reduces violations by over 50%, showcasing its ability to generalize the shield to similar/non-stationary environments despite subtle differences. This substantial reduction in violations, even in an unseen environment, highlights ADVICE's robustness and flexibility, which other methods lack. While ensuring safety, ADVICE achieves a comparable reward and goal-reaching frequency to the other methods, providing strong empirical evidence that its shield effectively supports transfer learning tasks.

## 5.3 CONSTRAINED ENVIRONMENT

While constrained MDP environments are less common in real-world scenarios compared to uncon-strained ones, to examine ADVICE's performance in such settings, we compared it against several methods in the Constrained Randomized Goal environment, as illustrated in Figure 5. Complete results are provided in Appendix C. Notably, ADVICE consistently maintains the lowest average episodic cost, outperforming both DDPG-Lag and CSC, which exhibit significant oscillations. These fluctuations reflect the inherent struggle of these methods to balance cost reduction with reward maximization, a common challenge highlighted in the literature for these methods (Liu et al., 2022). Interestingly, we did not observe such oscillations in environments with sparse costs, such as the unconstrained settings discussed in Section 5.1.

## 5.4 ADVICE ADAPTATION

To evaluate ADVICE's adaptation capabilities, we conducted a sensitivity analysis focusing on the impact of varying the distant history window $h_d$ and the recent history window $h_r$. For each $h_d$ and $h_r$ combination, Table 1 shows the number of consecutive episodes $K$ was fixed at a specific value, the frequency of $K$ adjustments, and the impact on mean safety violations.

Our findings reveal an impact of $h_r$ on the frequency of $K$ adjustments. A smaller $h_r$ leads to more frequent $K$ increases, allowing the system to quickly respond to immediate safety violations.

Table 1: Mean sensitivity analysis of $h_d$ and $h_r$ on the randomised goal environment for $E^{max} = 1000$

| $h_r$ | Metrics | $h_d$ | |
| | | 10 | 25 |
|---|---|---|---|
| 2 | Consecutive Episodes ($K = 3$) | $24.49 \pm 9.59$ | $31.09 \pm 4.53$ |
| | Consecutive Episodes ($K = 4$) | $1.00 \pm 0.00$ | $7.89 \pm 0.77$ |
| | Consecutive Episodes ($K = 5$) | $32.97 \pm 7.95$ | $151.60 \pm 3.09$ |
| | Changes of $K$ | $116.33 \pm 9.87$ | $27.67 \pm 4.62$ |
| | Safety Violations | $50.42 \pm 9.94$ | $57.88 \pm 7.64$ |
| 3 | Consecutive Episodes ($K = 3$) | $166.81 \pm 5.71$ | $117.35 \pm 8.07$ |
| | Consecutive Episodes ($K = 4$) | $21.10 \pm 6.86$ | $28.76 \pm 9.26$ |
| | Consecutive Episodes ($K = 5$) | $13.49 \pm 5.40$ | $149.68 \pm 8.71$ |
| | Changes of $K$ | $28.67 \pm 1.41$ | $19.33 \pm 2.83$ |
| | Safety Violations | $51.69 \pm 6.93$ | $54.45 \pm 5.10$ |
| 4 | Consecutive Episodes ($K = 3$) | $954.49 \pm 9.59$ | $169.07 \pm 9.02$ |
| | Consecutive Episodes ($K = 4$) | $22.60 \pm 6.71$ | $22.10 \pm 7.26$ |
| | Consecutive Episodes ($K = 5$) | $0.00 \pm 0.00$ | $181.35 \pm 4.92$ |
| | Changes of $K$ | $2.33 \pm 0.47$ | $17.77 \pm 2.72$ |
| | Safety Violations | $55.48 \pm 5.64$ | $50.17 \pm 5.47$ |

Conversely, a larger $h_r$ tends to stabilise $K$ by filtering out anomalies and adjusting only in response to sustained trends of increased violations. Similarly, $h_d$ influences the decrease of $K$; a smaller $h_d$ facilitates rapid decreases in $K$ following a reduction in safety violations, whereas a larger $h_d$ results in less frequent reductions, promoting stability in ADVICE's behaviour. The interaction between $h_d$ and $h_r$ minimally affects the overall rate of safety violations, suggesting that while these parameters impact the adaptiveness and stability of $K$, they do not directly correlate with safety violations. These insights highlight the role of $h_d$ and $h_r$ primarily as tuning parameters to balance responsiveness against stability in ADVICE.

## 5.5 Discussion

Despite ADVICE achieving a significantly lower safety violation rate compared to other methods, there are a few areas that warrant further exploration. Firstly, while ADVICE experiences a *cold-start* period due to the need to gather sufficient features for training the contrastive autoencoder, it still outperforms methods that start learning from timestep $t_0$ by significantly reducing violations. This challenge can be mitigated through transfer learning (as seen in Figure 4). Furthermore, although ADVICE increases computational demands due to continuous inference at each timestep, optimizing inference intervals or leveraging more efficient models can help balance performance and resource usage. This adaptation may help extend its applicability to resource-constrained settings without compromising safety. Lastly, in dynamic environments with temporally changing obstacles, ADVICE might benefit from incorporating temporal context through methods like LSTM, which would enhance its ability to handle these situations effectively. This could offer a promising avenue for further enhancing safety performance, despite the additional computational load.

## 6 Conclusion and Future Work

We introduced ADVICE, a post-shielding technique for the safe exploration of RL agents operating in complex *black-box* environments. ADVICE does not need *any* prior knowledge and uses a contrastive autoencoder to distinguish between safe and unsafe features efficiently. Our evaluation shows that ADVICE significantly reduces safety violations while maintaining competitive performance against state-of-the-art methods. Despite its effectiveness, ADVICE has areas for improvement, including mitigating the cold-start issue, reducing high computational demands, and addressing challenges with dynamic obstacles. In future work plan to explore meta-learning techniques (Hospedales et al., 2021) to address the cold-start problem, enabling faster activation of ADVICE without sacrificing performance. Quantisation or pruning could reduce computational demands, enhancing applicability in computationally constrained domains. We envisage that ADVICE serves as a foundational step in using neural network-based shielding for safe RL exploration in complex, black-box environments without *any* prior knowledge.

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

## A   Task Details

In the Safety Gymnasium (Ji et al., 2023a) [2] test-suite, a robot with Lidar sensors has to navigate through environments with obstacles to complete a given task. The test suite comes with a multitude of robots (e.g. Point, Ant, Car) and a set of tasks (e.g. Goal, Circle, Button) that can be evaluated. In our experiments, found in Section 5, we chose to use:

- **Car robot:** This robot has two wheels on the rear that the agent can control with one free-rolling front wheel. Steering and movement require nuanced coordination. The action space for the car is $[-1, 1]^2$, and the agent is shown in Figure 6.
- **Semi-random Goal**: A standard goal environment, where the agent aims to reach the goal at the end of the episode whilst navigating through six obstacles. The six obstacles have a static spawn, the agent and the goal have randomised positions every episode. We placed the six obstacles to form a large *wall*, where the agent can fit through to reach the goal but with an increased risk of crashing. In this instance, we want to determine if ADVICE and other safe RL exploration methods will learn to avoid the *wall* or risk going through it. Deployment trajectories in Figure 3, show the learnt trajectories of ADVICE and other methods.
- **Randomised Goal**: This environment is similar to the semi-random goal environment, with the additional complexity of the obstacles also having random spawns. This extra randomised aspect adds increased difficulty as the agent and safety mechanisms cannot memorise the positions of the obstacles to avoid.
- **Randomised Circle**: The agent has to circle in a given zone in this environment. The aim is to maximise speed and distance from the centre of the zone whilst navigating through three randomised obstacles. This scenario element further increases the task difficulty as now the obstacles to avoid are directly within the area where the agent can maximise its reward.
- **Constrained Randomised Goal**: This environment is similar to the random goal environment, but instead the task is set up as a constrained MDP. Instead of obstacles that terminate the episode, the task includes hazards that give the agent $-0.2$ cost per step when inside them. The agent cannot terminate in this task and instead has to minimise cost whilst maximizing rewards.

In all environments for all the tasks above, the agent uses *psuedo* Lidar to perceive objects in the environment. Each type of object (e.g. goal, obstacles) in the environment has its own separate Lidar observation, where a Lidar vector has 16 bins. All vectors are flattened into one observational vector and then given to the agent as the current state. For example, in the semi-random goal environment, there is a goal and a set of obstacles. Here the observational space is $[0, 1]^{32}$. All lidar vectors have a max distance of 3 meters. In both goal environments, an episode has a maximum timestep of 1000. In the circle environment, the maximum timestep is 500. A *goal reach* in both goal environments is defined as reaching the goal before the episode truncates, in the circle environment it is defined as being within the circle when the episode truncates. A *crash* is defined as the agent colliding with an obstacle, when this occurs, the episode is terminated shortly. Anything else is considered as the episode *timing out*.

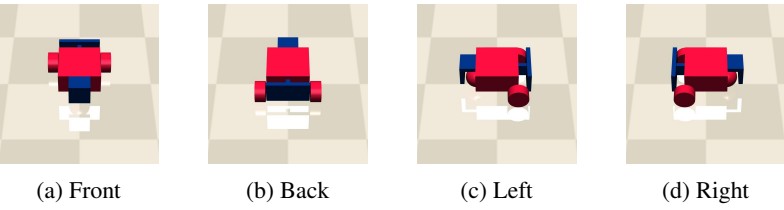

(a) Front          (b) Back          (c) Left          (d) Right

Figure 6: Different views of the Car robot in the Safety Gymnasium test suite (Ji et al., 2023b).

Each task has a separate reward function for the agent to maximise. Whenever the agent comes into contact with an obstacle, a constraint cost of $-1$ is given (the exception being the Constrained Randomised Goal task). The reward functions for each task are:

---

[2]More details here: https://github.com/PKU-Alignment/safety-gymnasium

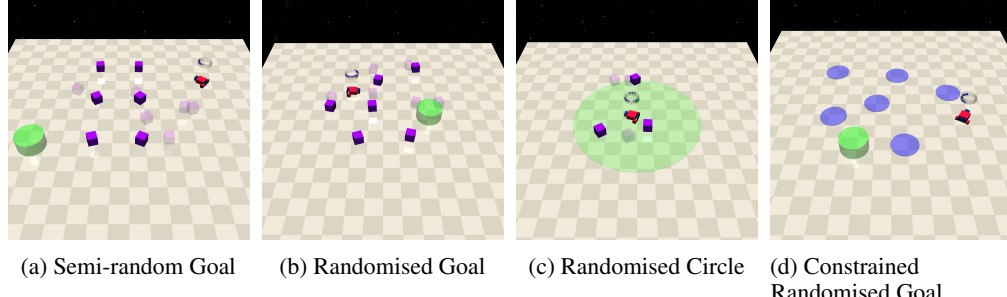

|  (a) Semi-random Goal | (b) Randomised Goal | (c) Randomised Circle | (d) Constrained Randomised Goal |

Figure 7: Example navigation tasks with varying complexity levels for evaluating ADVICE. The purple blocks are the obstacles, the green circles are the goals, and the red vehicle is the agent.

- **Semi-random, Randomised Goal & Constrained Randomised Goal**: $R_t = (D_{last} - D_{now})\beta$, where $D_{last}$ is the distance between the agent and the goal in timestep $t_{-1}$, $D_{now}$ is the distance between the agent and the goal in timestep $t$, and $\beta$ is the discount factor. Simply, the agent moving towards the goal, in terms of Euclidean distance, gains a positive reward. The agent moving away from the goal gains a negative reward. Reaching the goal gains a static reward of $+1$.
- **Randomised Circle**: $R_t = \frac{1}{1+|r_{agent}-r_{circle}|} * \frac{(-uy+vx)}{r_{agent}}$ where $(u, v)$ is the x-y velocity coordinates of the agent, $(x, y)$ are the x-y coordinates, $r_{agent}$ is the Euclidean distance of the agent from the origin of the circle, and $r_{circle}$ is the radius of the circle. Simply, the agent is rewarded for moving at speed along the circumference of the circle.

It should be emphasized that the agent or methods used in the evaluation have no prior knowledge of the task/environment/safety concern. Thus, we can define this environment and all tasks within as *black box*.

## B   THEORETICAL ANALYSIS

In this section, we theoretically analyze ADVICE and show that the expected probability of ADVICE misclassifying an unseen feature is bounded and can be decreased by diversifying the data collected before episode $E$.

**Theorem 1** *The probability of ADVICE misclassifying a feature is bounded by $exp(-\gamma/2\sigma^2)$, where $\gamma$ is the contrastive separation margin and $\sigma^2$ is the variance of the assumed Gaussian noise in the latent space.*

The contrastive separation margin in the latent space is defined as:

$$\gamma = \min_{f_s \in S, f_u \in U} \|\text{ENCODE}(f_s) - \text{ENCODE}(f_u)\|_2 \tag{8}$$

The noise in the latent space is assumed to follow a Gaussian distribution $\epsilon \sim N(0, \sigma^2)$. In ADVICE, an unseen feature $f$ is classified as safe if the K-nearest neighbours of $\text{ENCODE}(f)$ contain safer than unsafe features. So, let $d_s = \|\text{ENCODE}(f) - \text{ENCODE}(f_s)\|_2$ and $d_u = \|\text{ENCODE}(f) - \text{ENCODE}(f_u)\|_2$ define the Euclidean distance to the nearest safe and unsafe feature for an unseen feature $f$. Therefore, misclassification occurs when $f \in S$ and $d_u < d_s$ or $f \in U$ and $d_u > d_s$. The contrastive separation margin $\gamma$ ensures that, in a noise-free case $\|\text{ENCODE}(f) - \text{ENCODE}(f_s)\|_2 \geqslant \gamma$. In a realistic presence of noise, the distances $d_s$ and $d_u$ are perturbed by $\epsilon_s, \epsilon_u \sim N(0, \sigma^2)$. Therefore:

$$d_u - d_s = \|\text{ENCODE}(f) - \text{ENCODE}(f_u)\|_2 - \|\text{ENCODE}(f) - \text{ENCODE}(f_s)\|_2 \approx \gamma + \epsilon \tag{9}$$

where $\epsilon = \epsilon_u - \epsilon_s \sim N(0, \sigma^2)$ and is independent and Gaussian. The probability of $d_u < d_s$ (misclassification) when $f \in S$ is given by:

$$P(d_u < d_s) = P(\gamma + \epsilon < 0) = P(\epsilon < -\gamma) \tag{10}$$

Since $\epsilon \sim N(0, \sigma^2)$, we can normalise it so that:

$$P(d_u < d_s) = P\left(Z < -\frac{\gamma}{\sqrt{2} \cdot \sigma}\right) \tag{11}$$

where $Z \sim N(0, 1)$. Using the cumulative distribution function of the standard normal distribution $\Phi$, we get:

$$P(d_u < d_s) = \Phi\left(-\frac{\gamma}{\sqrt{2} \cdot \sigma}\right) \tag{12}$$

ADVICE uses K-nearest neighbours to classify an unseen feature $f$. If $\gamma$ is large relative to $\sigma$, the probability of misclassifying an unseen feature decreases exponentially. So, we can define the expected number of misclassified features to be:

$$E[misclassified\ features] \leqslant N \cdot exp\left(-\frac{\gamma}{2\sigma^2}\right) \tag{13}$$

The noise $\sigma$ in the latent space can come from: noisy data, imperfect model training, randomness in batch sampling, etc.

**Theorem 2** *The probability of ADVICE misclassifying a feature decreases exponentially with improved data diversity, bounded by $exp(\sqrt{H(F_E)}/2\sigma^2)$.*

Let $\gamma_m$ express the effective achieved separation margin between sets $S$ and $U$, where $\gamma_m \leqslant \gamma$. Equality only holds under ideal conditions, such as perfectly diverse data, perfect model training, no data noise, etc. The diversity of the feature set $F_E$ collected before episode $E$ can be quantified using entropy:

$$H(F_E) = -\sum_{f \in F_E} p(f)\ log\ p(f) \tag{14}$$

where $p(f)$ is the probability distribution of features $f \in F_E$. Higher entropy corresponds to a broader set of features, ensuring greater diversity. Greater diversity results in more representative embeddings, given good model training, allowing the contrastive loss function to achieve better separation and clusterings of sets $S$ and $U$. The effective separation margin $\gamma_m$ depends on $H(F_E)$. As diversity increases, the embeddings for $S$ and $U$ become more separable, thereby $\gamma_m \propto k \cdot \sqrt{H(F_E)}$ where $k > 0$ is a proportionality factor that links the entropy $H(F_E)$ of the feature set to the effective separation margin $\gamma_m$. It encapsulates the influence of latent space geometry, scaling properties, and model-specific parameters. While $k$ may vary depending on the training process and feature distribution, it is assumed to be stable for a given setup. Empirically, $k$ can be estimated by observing the relationship between $\gamma_m$ and $H(F_E)$ across diverse datasets or configurations. From Theorem 1, with the substitution of $\gamma_m$ for $\gamma$, the probability of misclassifying a feature is bounded by:

$$\begin{aligned} P(misclassification) &\leqslant \exp\left(-\frac{\gamma_m}{2\sigma^2}\right) \\ &\leqslant \exp\left(-\frac{k \cdot \sqrt{H(F_E)}}{2\sigma^2}\right) \end{aligned} \tag{15}$$

therefore showing that increasing the diversity of the feature set reduces the misclassification probability of a feature exponentially.

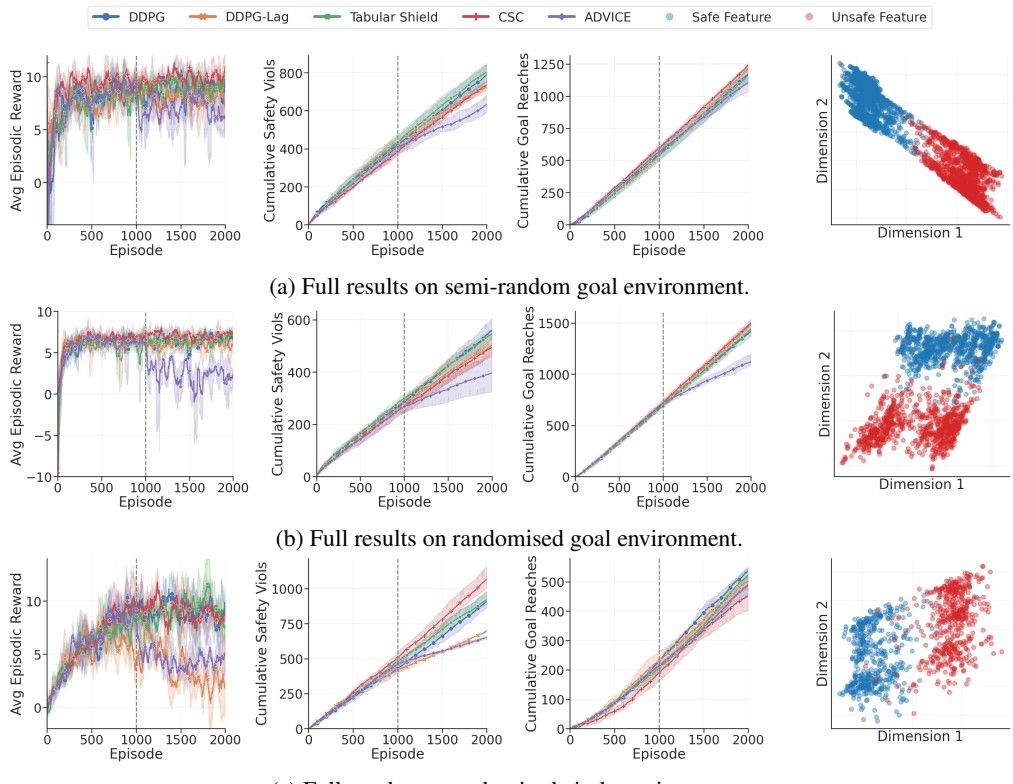

(a) Full results on semi-random goal environment.

(b) Full results on randomised goal environment.

(c) Full results on randomised circle environment.

Figure 8: Average episodic reward, cumulative safety violations, cumulative goal reaches of examined methods (DDPG, DDPG-Lag, Tabular shield, Conservative Safety Critic, ADVICE) and example latent space visualisation for the semi-random goal (top), randomised goal (middle) and randomised circle (bottom) environments.

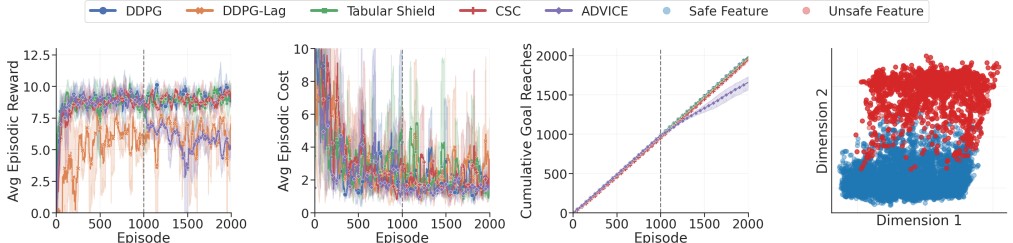

Figure 9: Average episodic reward, average episodic cost, cumulative goal reaches, and example latent space visualisation on the constrained randomised goal environment.

## C  FULL TRAINING RESULTS

In Section 5.1, we show the main results for all methods in a set of tasks. For fair comparison, we show results from episode 1000 and standardise all metrics to zero. Below, in Figure 8, we show the unstandardised results for the same experiments.

In all experiments, the Tabular Shield method performs approximately the same as the standard DDPG agent. To show why this behaviour occurs, we plot the average shield activations for ADVICE and the Tabular Shield in Figure 10. From these results, it is evident that the Tabular Shield does not once activate during training across all tasks. This is due to the high dimensionality of the environments evaluated. Even though the features stored are discretised to 1 decimal place, the agent has to observe the exact same values across all $\approx 32$ dimensions plus the actions for the shield to

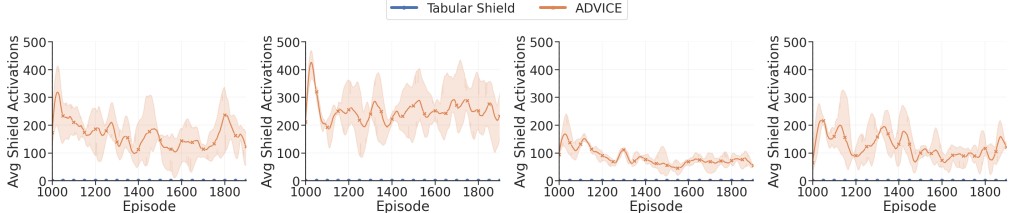

Figure 10: The average shield activations for ADVICE and the Tabular Shield in the semi-random goal, random goal, random circle, and constrained random goal environments respectively.

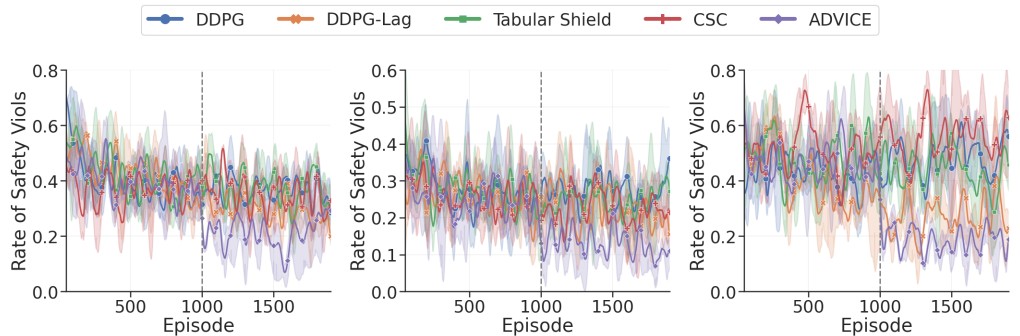

Figure 11: The rate of safety violations for all methods in the semi-random goal, random goal, and circle environments respectively.

activate. Our experiments show that this method fails in these types of environments. A trend that can be noticed with ADVICE is when the shield is first activated, the amount of interventions starts relatively high. As training progresses this number reduces, which shows that the agent learns to adapt to the shield's understanding of safety.

Figure 11 shows the rate of safety violations during training. This outcome further validates the results and conclusions discussed in Section 5.1. We observe that the DDPG and Tabular Shield agents perform similarly. The CSC agent, due to sparse data, underestimates safety and only reduces violations by a fractional amount. The DDPG-Lag agent manages to reduce the safety violations somewhat towards the end of training, which is particularly evident in the circle environment. Once ADVICE is turned on, it significantly reduces the rate of safety violations in all environments.

# D  PARAMETER ANALYSIS

In this section, we present an extended analysis of ADVICE to display the robustness and adaptability of the approach. These experiments were chosen to explore the effects of varying $K$ thresholds, and the timing of ADVICE's activations $E$.

A user can specify the conservativeness of ADVICE using the safety threshold $K$. In Figure 12, we evaluate how this parameter affects the model's safety, and performance. The results are clear, increasing $K$ leads to a more conservative behaviour as hypothesised. The reward decreases a small amount as well as the cumulative goal reaches, however, it also results in fewer safety violations. Conversely, decreasing $K$ allows the underlying DDPG agent more freedom. As a result, average reward and goal reaches are increased at the expense of safety violations. These findings display a clear trade-off between return efficiency and safety assurance.

ADVICE has a *cold-start*, meaning it requires some period of time before activation to collect data in order to work efficiently. We acknowledge that this can affect the performance of ADVICE evidently we show the results of various activation points in Figure 13. To allow for a fair comparison as possible, we show the rate of safety violations for the subsequent 1000 episodes after activation. Again, we visualise a trade-off. Delaying ADVICE's activation for longer results in fewer safety violations and increased goal reaches. However, the RL agent observes more cumulative safety

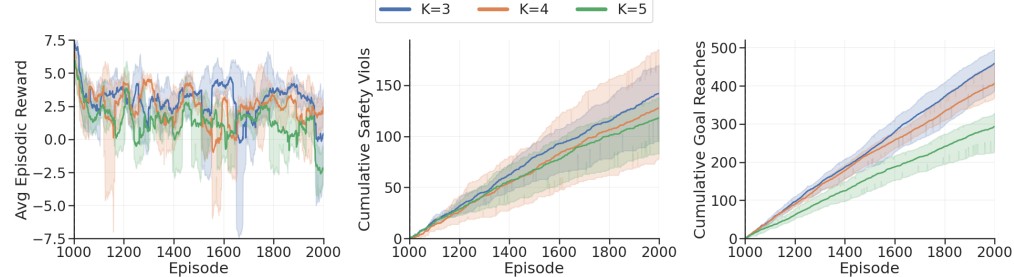

Figure 12: Average episodic reward, cumulative safety violations, and cumulative goal reaches of various values of $K$ on the randomised goal environment.

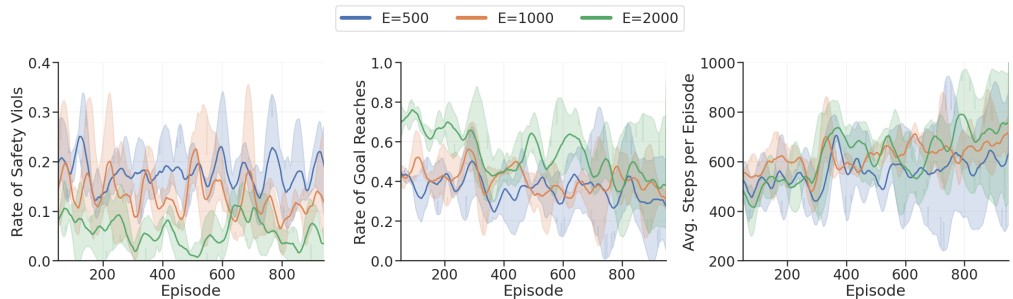

Figure 13: The rate of safety violations, and rate of goal reaches when ADVICE is activated at different intervals $E$.

violations up to the point of activation. Starting ADVICE earlier decreases the number of safety violations up to activation but gives the autoencoder fewer data points to train off. Evidently, safety violations are not reduced to the same magnitude and goal reaches also decrease. This is to be expected with any neural network-based approach.

Based on results in Figure 2b, we hypothesise that the reduced reward and cumulative goal reaches is a result of ADVICE not having *enough* time to complete the task. As seen in Figure 3, ADVICE learns to take a longer route to the objective, so by doubling the maximum step count allowed per episode, we expect to see an increase in cumulative goal reaches, average reward, and no increase to safety violations. Results for this experiment are shown in Figure 14. As expected, given more time to complete the task, ADVICE now reaches the goal more than when the maximum step counter is the default value. As a result, we observe an increase in average episodic return much closer to baseline methods.

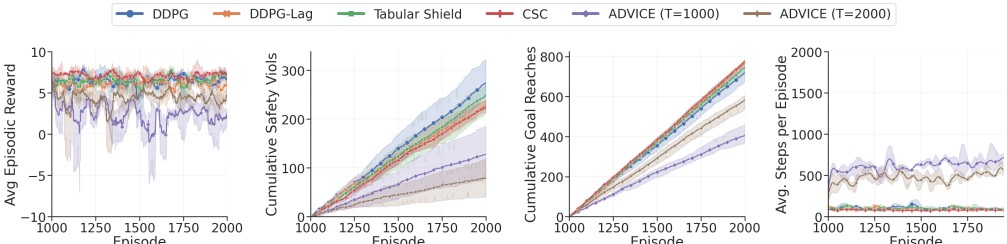

Figure 14: Average episodic reward, cumulative safety violations, and cumulative goal reaches of various methods on the randomised goal environment where the maximum episodic steps are doubled.

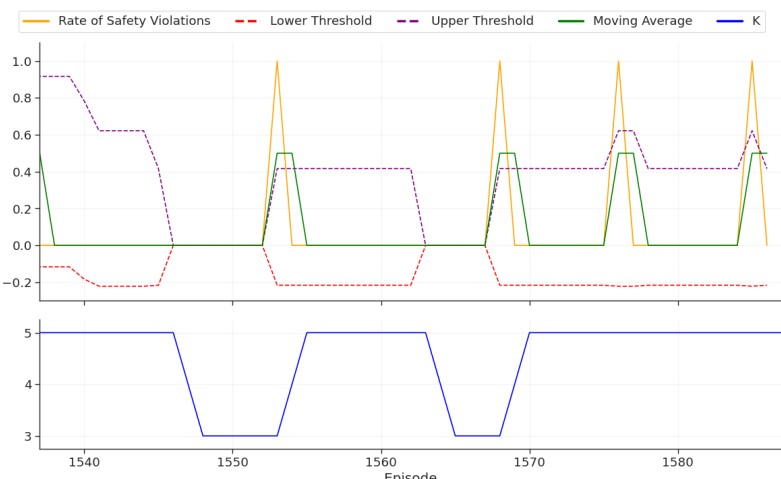

Figure 15: The rate of safety violations, lower threshold $(MA_{h_d} - \sigma_{h_d})$, upper threshold $(MA_{h_d} + \sigma_{h_d})$, recent moving average $(MA_{h_r})$, and value of $K$ during an example run where $h_d = 10$ and $h_r = 2$.

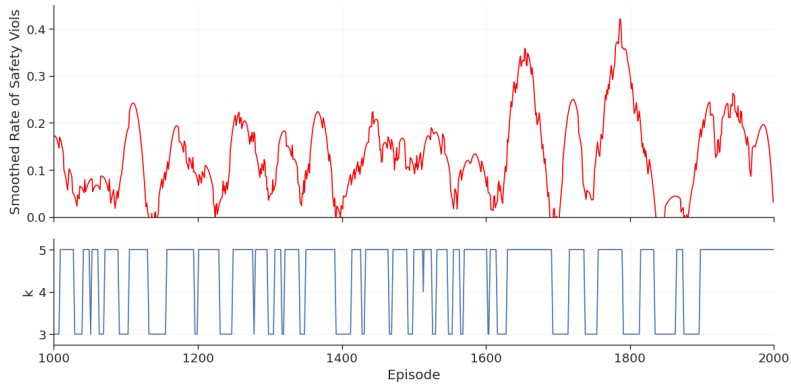

Figure 16: Rate of safety violations and value of $K$ on an example run (one random seed) that shows the adaptation of ADVICE ($h_d = 10, h_r = 2$).

# E   ADAPTIVE ADVICE

To validate that Adaptive ADVICE correctly increases and lowers $K$ during training, we plot an example visualisation window in Figure 15 showing the rate of safety violations, the upper $(MA_{h_d} + \sigma_{h_d})$ and lower $(MA_{h_d} - \sigma_{h_d})$ thresholds, the moving average $(MA_{h_r})$, and the value of $K$.

It can be seen that when the recent moving average $(MA_{h_r})$ is above the upper threshold $(MA_{h_d} + \sigma_{h_d})$, the adaptive module correctly increments $K$. An example of this can be seen at episode $1553$. The agent crashes, and both thresholds adjust but the recent moving average climbs above the upper threshold, increasing $K$ as a result. In subsequent episodes afterwards, the recent moving average falls between both thresholds. Here the adaptive module correctly keeps $K$ at the same value until episode $1564$ where the moving average is equal to the lower threshold. As a result, $K$ is decreased. This example window validates that the adaptive shield works as expected and also provides an insight into how it works during training.

Table 2: Summary of hyperparameters in the DDPG algorithm and the ADVICE shield.

| Parameter | DDPG | Parameter | ADVICE Shield |
|---|---|---|---|
| Network size | (256, 256) | Size of network | (512, 2, 512) |
| Optimizer | Adam | Optimizer | NAdam |
| Actor learning rate | 2e-3 | Learning rate | Reduce on plateau |
| Critic learning rate | 1e-3 | Batch size | 32 |
| Size of replay buffer | 2e5 | Max epochs | 500 |
| Batch size | 64 | No. Neighbours ($K^{max}$) | 5 |
| Gamma | 0.95 | No. Safe neighbours ($K$) | 4 |
| Tau | 5e-3 | Losses | (MSE, MSE, CL) |
| Ornstein-Uhlenbeck noise | 0.2 | Loss weights | (1, 1, 1.25) |
| - | - | Unshielded Episodes ($E$) | 1000 |

# F HYPERPARAMETER ANALYSIS AND COMPUTATIONAL OVERHEADS

This section lists the hyperparameters used by all models and ADVICE. Table 2 summarises all hyperparameters used in Section 5. We will refer the reader to our source code repository [1] for the remaining details.

Using all model configurations in Table 2, a single ADVICE run (one random seed) takes 12, 24, 12, and 12 hours of training, respectively, in the semi-random goal, random goal, random circle, and constrained random goal environments. For all other methods, a single run takes 3, 4, 3, and 5 hours in the same environments, respectively. All experiments were run on a large computing cluster utilising two Nvidia H100 GPUs, 16 CPUs, and up to 500GB memory.

Hyperparameters for the DDPG algorithm started with author recommendations (Lillicrap et al., 2015). They were manually tuned afterwards to achieve a high performance on individual environments before tests were carried out, meaning the RL algorithm for all approaches was of high performance and fair comparison. Hyperparameters for ADVICE were manually tuned for performance in CL loss and MSE loss. Some hyperparameter analysis was conducted in Section D to justify certain choices. Parameters for the DDPG-Lag method started with recommendations (Stooke et al., 2020) and were tuned for performance in our experiments.

