# OpenReview forum: "Safe Reinforcement Learning in Black-Box Environments via Adaptive Shielding"
_ICLR.cc/2025/Conference — Submitted to ICLR 2025_

### Official Review · Reviewer_Ff6i · 2024-10-29

**Soundness:** 3
**Presentation:** 2
**Contribution:** 2
**Rating:** 3
**Confidence:** 4

**Summary:**

The paper presents a framework to address the problem of safety within unknown environments using a combination of techniques including shielding, contrastive learning and latent feature representations. The framework therefore consists of many components that have been combined under one roof. The target is so called black box environments where the benefit of information relating to safe states is not a priori available so special techniques need to be employed to enable training and execution to proceed as safely as possible. The empirical results show that the framework, ADVICE increases safety violations by a meaningful margin though often at the expense of return

**Strengths:**

The methodology is intuitive and the motivation and requirement of each component are reasonably well explained. The problem setting is one of significant interest to the RL community and the approach seems original.

**Weaknesses:**

Although some parts of the paper are well explained, especially the non-technical elements and the intuition of the algorithm, many important details are skipped. This leaves the reader to simply guess how important aspects of the framework work. Also, some maths expressions seem to have been written without due care which further adds to confusion. The framework involves various interdependent learning processes - specifically, in addition to the regular policy updates, the action and hence the reward received at a given state depends on the latent representation (which presumably is being updated) as well as the value of K which varies in quite an abrupt manner. This generates concerns about convergence guarantees and the numerical stability of the framework. The paper would benefit from some analysis or at least a discussion on this aspect. Additionally, within the framework there are a number of functions within the method that have max/min functions - this introduces a high level of discontinuity which can be harmful for some learning methods e.g. gradient-based learning methods, and may cause numerical instabilities.

The novelty of the paper as a whole could be more clearly spelled out. Lastly, although the framework achieves a significant reduction in safety, this often comes at a notable expense to the return. I would like to see how this compares to the frameworks that have some calibration aspect for the performance:safety ratio.

**Questions:**

1. In environments with a highly non-smooth reward function, similar features can induce very different rewards (specifically different levels of safety). What exactly is the function $h_\theta$ in (1)? If this function does not relate to safety or reward then the issue of non-smoothness in the safety w.r.t. features may be problematic here.

2. Given that $\mathcal{U},\mathcal{S},\mathcal{I}$ are defined as sets, the construction of $g$ in (3) does not seem like a function since it maps elements of $F_E$ to sets. Perhaps $\mathcal{U},\mathcal{S},\mathcal{I}$ need not be sets for this construction.

3. The construction of $\mathcal{C}$ in (4) also seems spurious since it maps a product of sets to a set which is the union of other sets. Also, it seems not to consider the cases of two dissimilar features in $U$ and secondly, two dissimilar features in $S.

4. What are $S$ and $U$ in (4) - in page 3, the subindex of $f$ represents time, the indices $i,j \in S$ etc. in (4) therefore seem incorrect.

5. In relation to the construction of the function $g$ in (3), in the black box setting, how can we know which are accepting states?

6. In line 6 of the algorithm, how can we be sure that constructing an action by taking a cartesian product across action dimensions will produce a valid action in the environment? The set of valid actions may only be a subset of this product space e.g. a robot may not be able to lift both its feet off the ground simultaneously.

7. What is the definition of a safety violation - specifically, is there a notion of safety violation that can be used to detect such violations in black-box environments?

---

> ### Author Response · Authors · 2024-11-19
>
> Dear reviewer,
> Thank you very much for your valuable review, comments regarding significance and novelty, and questions regarding ADVICE! We have integrated some of your points regarding clarity into an updated version of our submission. Please find answers to your questions below:
>
> ## Questions and Answers
>
> > In environments with a highly non-smooth reward function, similar features can induce very different rewards (specifically different levels of safety). What exactly is the function in (1)? If this function does not relate to safety or reward then the issue of non-smoothness in the safety w.r.t. features may be problematic here.
>
> A: Equation (1) defines the contrastive loss function used to train the autoencoder within ADVICE. ADVICE does not directly incorporate the rewards/reward function within its method. We use the contrastive loss function to identify similar/dissimilar characteristics between safe and unsafe data in a self-supervised manner. Therefore, variations in reward smoothness are unlikely to impact ADVICE’s ability to classify data. Contrastive loss mitigates non-smoothness with regard to features by focusing on clustering safe and unsafe features separately based on their safety labels rather than relying on smooth transitions in feature space [1]. This contrastive loss characteristic allows ADVICE to handle abrupt shifts in safety effectively, reinforcing boundaries between safe and unsafe regions (see Figure 2).
>
> [1] Hadsell, R., Chopra, S. and LeCun, Y., 2006, June. Dimensionality reduction by learning an invariant mapping. In 2006 IEEE computer society conference on computer vision and pattern recognition (CVPR'06) (Vol. 2, pp. 1735-1742). IEEE.
>
> > Given that U, S, and I are defined as sets, the construction of g in (3) does not seem like a function since it maps elements of F_{E} to sets. Perhaps U, S, and I need not be sets for this construction.
>
> A: Thank you for bringing this incorrect formulation to our attention. We have amended Equation 3 to prevent the formulation of U, S, and I from being both subsets of F_{E} and labels.
>
> > The construction of C in (4) also seems spurious since it maps a product of sets to a set which is the union of other sets. Also, it seems not to consider the cases of two dissimilar features in U and secondly, two dissimilar features in S.
>
> A: Following from the previous point, we have amended the formulation issue with Equation (4). Again, we thank you for bringing this to our attention and apologise for any confusion. For dissimilar features within the same category (e.g., U or S), the contrastive loss function enables the model to identify similarities within each category and differences across categories, ensuring effective learning regardless of intra-category dissimilarities.
>
> > What are S and U in (4) - in page 3, the subindex of f represents time, the indices i, j in S etc. in (4) therefore seem incorrect.
>
> A: In Equation (4), S and U represent subsets of safe and unsafe features from F_{E}, respectively, and are indexed by i and j to denote pairwise elements within each category for contrastive loss. We agree that clarifying the notation could help avoid confusion with the temporal subindex of f on page 3. To mitigate this, Equation 4 has been amended to remove the inconsistent subscript.
>
> > In relation to the construction of the function g in (3), in the black box setting, how can we know which are accepting states?
>
> A: Even in black-box environments, accepting states can be recognized when encountered, as they typically represent clear outcomes (e.g., goal achievement or task completion) identifiable from the environment’s feedback. This characteristic allows us to label these states accordingly, ensuring that g in Equation 3 accurately classifies them.
>
> > In line 6 of the algorithm, how can we be sure that constructing an action by taking a cartesian product across action dimensions will produce a valid action in the environment?
>
> A: In line 6 of Algorithm 1, the Cartesian product is only applied within the valid action space, ensuring that all generated actions remain within the bounds and constraints of the environment. This prevents ADVICE from producing infeasible actions, such as those that violate physical limitations. We have revised line 241 to explicitly state that ADVICE generates a set of valid candidates.
>
> > What is the definition of a safety violation - specifically, is there a notion of safety violation that can be used to detect such violations in black-box environments?
>
> A: In black-box environments, a safety violation is represented by a terminal state which can be recognized when encountered in an MDP. This is similar to the question regarding accepting states above. We have improved the clarity of Line 311 based on this point.
>
> **Questions and Answers continue in the next comment...**

---

> > ### Author Response · Authors · 2024-11-19
> >
> > **Questions and answers continued from the previous comment...**
> >
> > > The novelty of the paper as a whole could be more clearly spelt out.
> >
> > A: We emphasize ADVICE’s novelty in multiple sections, notably in lines 52–64 and line 538. ADVICE represents a novel post-shielding approach that enables RL agents to explore safely within complex black-box environments without any prior knowledge.  ADVICE introduces several innovations, including a contrastive neural network shield, a new definition for recognizing safety categories based on terminal and accepting states, and an adaptive mechanism for adjusting conservatism based on recent performance, which enhances its robustness across various tasks. Together, these contributions position ADVICE as a foundational method for neural network-based shielding in black-box environments.
> >
> > > Although the framework achieves a significant reduction in safety, this often comes at a notable expense to the return. I would like to see how this compares to the frameworks that have some calibration aspect for the performance: safety ratio.
> >
> > A: We acknowledge that certain frameworks may employ a "safety ratio" or similar calibration metrics to balance safety and return. While ADVICE does not explicitly include such a ratio, it achieves strong safety performance while maintaining competitive returns relative to other methods (see Figures 2, 4 and 5). A comparison with frameworks incorporating safety-performance calibration is a valuable direction which we plan to explore in future work.
> >
> > > Within the framework there are several functions within the method that have max/min functions - this introduces a high level of discontinuity which can be harmful for some learning methods e.g. gradient-based learning methods, and may cause numerical instabilities.
> >
> > A: While ADVICE’s adaptation mechanism does include min/max operations to control the safety threshold K, Equation (5), these operations are applied outside of the gradient computation and are not part of the contrastive autoencoder’s learning process. We did not observe any numerical instabilities in our experiments, as these operations simply act to limit K within practical bounds, ensuring stability in adaptive behaviour rather than influencing gradient-based updates directly.
> >
> > > The framework’s interdependent learning processes, including policy updates, latent representation updates, and abrupt variations in K, raise concerns about convergence guarantees and numerical stability. Some analysis or discussion on this would strengthen the paper.
> >
> > A: In ADVICE, the latent representation learning is designed to classify safety features independently of immediate policy updates, mitigating instability from interdependent processes. The parameter K is adjusted gradually using moving averages of safety violations, ensuring smoother transitions; see Equations (5)-(7). Additionally, all updates occur within bounded values, minimizing abrupt shifts. Empirically, we observed no instability in our experiments, as the adaptive mechanisms allow ADVICE to handle varying safety requirements effectively while maintaining stability; see Figure 12 that shows results for K = {3, 4, 5}.

---

> ### Comment · Reviewer_Ff6i · 2024-11-21
>
> Thanks to the authors for their responses. However, some of my questions remain.
>
> 1. Your explanation makes sense to me however in the absence of a concrete example of $h_\theta$, it is still difficult to know how the claims you make can be realised in practice. Can you specify an example of the function $h_\theta$?
>
> 5/7. To evaluate the functions $g$ and $\mathcal{S}$ at all states, to me it seems that each state will need to have been encountered at least once therefore committing the agent to unsafe state visitations. Can you clarify if this is the case or does the method have a technique that enables the output of these functions to be predicted given visits to other states?
>
> 6. My point here is that the space of valid actions may not be closed under the Cartesian product operation. As in my example, a robot lifting either foot off the ground (and keeping other foot grounded) may be a valid action but lifting both feet therefore performing both of these individually valid operations simultaneously may be an invalid operation.

---

> > ### Author Response · Authors · 2024-11-22
> >
> > We thank you for your discussion so far, it has been extremely helpful and raised some astute points. Please find the responses to your further questions below:
> >
> > ## Questions and Answers
> >
> > > Your explanation makes sense to me however in the absence of a concrete example of h_theta, it is still difficult to know how the claims you make can be realised in practice. Can you specify an example of the function h_theta?
> >
> > A: Within ADVICE, the function $h_\theta$ is the encoder part within an encoder-decoder neural network (see Advice Construction step in Figure 1). Specifically, $h_{\theta}$ maps input features (in ADVICE this is state-action pairs) into a latent embedding. As an example from our code repo, please see a snippet of the encoder network $h_\theta$ below.
> >
> > ```python
> > def get_encoder(self):
> >     inputs = layers.Input(shape=(input_shape,))
> >
> >     x = layers.Dense(512, kernel_initializer=...)(inputs)
> >     x = layers.LeakyReLU()(x)
> >
> >     ....
> >
> >     latent = layers.Dense(latent_dim, activation='linear')(x)
> >     model = tf.keras.Model(inputs, latent, name='encoder')
> >     return model
> >
> > ```
> > and perform inference on this network to receive the embeddings, we can do the following (see line 14 from Algorithm 1):
> >
> > ```python
> > self.encoder.predict(features)
> >
> > ```
> >
> >
> > > To evaluate the functions g and S, at all states, to me it seems that each state will need to have been encountered at least once therefore committing the agent to unsafe state visitations. Can you clarify if this is the case or does the method have a technique that enables the output of these functions to be predicted given visits to other states?
> >
> > A: The agent must encounter some safe and unsafe states during the initial unshielded exploration phase, but it does not entail that each state will need to have been encountered at least once therefore committing the agent to unsafe state visitations (i.e., there is no need to visit all possible states). The purpose of the contrastive autoencoder (CA) is to generalise from a subset of visited features by learning what safe features have in common (e.g., away from obstacles or near the goal) and how they differ from unsafe features (e.g., close to obstacles). Function $g$ only labels features as safe, unsafe, or inconclusive based on the outcomes of these interactions. The set $S$ then consists of features that have been labelled as safe by using the function $g$. Using these labelled features, the CA constructs a latent space where safe and unsafe features are well separated (see learnt embeddings in Figure 1 and right-most column in Figure 2), enabling ADVICE to classify unseen states and actions without requiring the agent to encounter them directly.
> >
> > > My point here is that the space of valid actions may not be closed under the Cartesian product operation. As in my example, a robot lifting either foot off the ground (and keeping the other foot grounded) may be a valid action but lifting both feet therefore performing both of these individually valid operations simultaneously may be an invalid operation.
> >
> > A: Thanks for the follow-up question on this point. In general, the space of valid actions may indeed not be closed under the Cartesian product operation, such as your example regarding the robot lifting its feet. In ADVICE, we address this implicitly through a validity check after constructing the Cartesian product, which considers mission-specific constraints and retains only the valid action space. We added a small clarification in the paper (line 244) to make this explicit.

---

> > > ### Author Response · Authors · 2024-11-25
> > >
> > > Dear Reviewer Ff6i,
> > >
> > > As the rebuttal deadline approaches, we hope that our responses have satisfactorily addressed all of your concerns.
> > >
> > > Should you need further clarification or additional explanations, we would be more than happy to provide them.
> > >
> > > Thank you for your time and consideration.
> > >
> > > Sincerely,
> > >
> > > ADVICE Authors

---

### Official Review · Reviewer_C6Rd · 2024-10-31

**Soundness:** 2
**Presentation:** 2
**Contribution:** 2
**Rating:** 3
**Confidence:** 4

**Summary:**

This paper proposes a post-shielding method ADVICE to reduce the constraint violation during the safe RL training. It first trains a neighbor model to classify the safety of state-action pair by contrastive learning in embedding space. Then it leverages this model to identify the safety of new state-action by the statistics of its K nearest neighbors and corrects the unsafe action to a safe one from a selected safe set. The authors conduct experiments on several safety gymnasium environments, which show that the proposed method can reduce the cumulative safety constraint violation during training.

**Strengths:**

- This paper proposes a new shield-based method for safe exploration, which is an important problem for application of RL.
- Overall, the paper is clearly written (e.g., fig. 1) and easy to follow.
- The idea of classifying the safety of state-action in latent space is novel.

**Weaknesses:**

- My biggest concern is the effectiveness of neighbor model in step 1, which determines whether a new state-action is safe or not. However, this key component in the proposed shielding method is trained based on data collected in an initial unshielded stage (line 161). During the execution, the policy will be updated and differ from the initial policy, which will lead to a severe distribution shift of state-action pair. Therefore, it's very questionable whether the neighbor model can still distinguish safe state from the unsafe one given unseen new state-action distribution.
- The construction of safe or unsafe sets $\mathcal S, \mathcal U$ is also problematic. In eq. (3), the feature (state-action pair) is classified as safe if it's start or next state reaches the goal while the feature with next state crashed is unsafe. Such classification obviously introduces some spurious correlations, e.g., the state-action pair is identified as unsafe because it's far from the goal instead of it will truly crash with obstacles. Meanwhile, the contrastive learning objective excludes the inconclusive features, the majority of the collected data, which makes the coverage of training data on state-action space very limited and further exacerbates the distribution shift issue.
- In experiment, although the results in fig.2 show that ADVICE has smaller cumulative safety violation, ADVICE also performs similarly to baselines on some other tasks (e.g., fig. 8(c) &9). Meanwhile, it's very weird that DDPG-lag and DDPG have very close results (also see questions), suggesting the DDPG-lag baseline is not well-tuned.

Minor issues:
- line 142, distinguish -> distinguishes
- Eq.(2), $F$ -> $\mathcal F$
- many notations are not defined before using. E.g., $\mathbb Z_+$ in eq. (2), $E$ in line 2 of Algorithm 1.

**Questions:**

- What are the data of visualization in last column in fig.2? Are they training data for neighbor model?
- Does the "constrained randomised goal" correspond to "randomised CMDP" in fig.7(d)?
- Why do not you use the standard tasks provided by safety-gymnasium (e.g., PointGoal, CarButton, ...) instead of customizing the tasks? The original tasks should be harder because all the layouts (agent, goal and obstacles) are randomized in each episode and they can better test the ability of safety exploration.
- In the experiment, why are the performances of DDPG and DDPG-lag very similar in terms of reward or safety violation? DDPG does not take safety into the consideration and only aims to maximize the reward. The lagrangian in DDPG-lag will thus be almost useless if it performs similarly to DDPG.

---

> ### Author Response · Authors · 2024-11-18
>
> Dear reviewer,
> Thank you very much for your valuable review, comments about ADVICE’s novelty, and fruitful comments regarding our paper! Please find answers to your questions below:
>
> ## Questions and Answers
>
> > I question the neighbour model's effectiveness in distinguishing safe from unsafe state-action pairs due to potential distribution shifts caused by policy updates during execution.
>
> A: This is a valid concern, and we thank the reviewer for this question. Indeed, distribution shift can occur as the policy evolves. However, we explicitly consider this scenario with the randomised environments used for evaluating ADVICE (Section 5). In particular, in the randomised goal, randomised circle and constrained randomised goal environments (lines 314-319), the following environment information is randomised per episode: the position of obstacles, the agent’s position and its rotation, and the position of the goal. Accordingly, the RL agent was encouraged to account for this type of distribution shift during training and consequently to cater for distribution shifts due to policy updates. Furthermore, our empirical results in the transfer learning experiment (Section 5.2) demonstrate that ADVICE remains effective even when the policy deviates from the initial data collection stage. Notably, the transfer learning experiment (Figure 4) shows that ADVICE trained on one distribution and then deployed in a different environment continues to distinguish safe from unsafe features successfully. This concrete result suggests that combining the neighbours model and autoencoder yields an effective ADVICE method capable of generalizing well despite distribution changes. However, analyzing the robustness of the neighbour's model will be a very interesting avenue for future work that we definitely will look to.
>
> >The construction of safe or unsafe sets is also problematic. This can introduce spurious correlations.
>
> A: We acknowledge the potential for spurious correlations between, and this is a potential common comment for methods that utilise deep learning. We can only say that our empirical results demonstrate ADVICE’s effectiveness compared to other state-of-the-art methods (Figures 2 and 4). As long as the data is diverse (see Theorem 2 in Appendix B), these negative correlations can be reduced. For example, in the randomised goal, unsafe features (failures) can be seen close to the goal and away. This directs the model to look away from the goal distance itself and towards the obstacle distance. Theorem 2 in Appendix B provides theoretical results regarding data diversity and how it affects ADVICE’s effectiveness.
>
> > The contrastive learning objective excludes the inconclusive features.
>
> A: In black-box environments, limited confirmed information can be derived about absolute safety, i.e., conclusive information that the state-action pair is safe or not. A key contribution to the paper was the classification of safe and unsafe features using accepting and terminal states, facts that we have derived about finite-horizon MDPs that we can leverage for safe exploration. Inconclusive features lack a clear safety classification, and such features have the potential to pollute the dataset and yield ineffective shields. Including these features would introduce significant ambiguity into the ADVICE method (as an inconclusive feature could lead to an accepting/terminal state in the future with no direct separation between the two). Future work would consider exploring these inconclusive features as if some knowledge can be derived from them, it would further improve ADVICE. However, this is a significant challenge within itself, and due to ambiguity, we focus on the conclusive features that can be confirmed as either safe or unsafe. Our experimental results demonstrate that even without using the inconclusive features, ADVICE can produce an effective shield that outperforms the state-of-the-art approaches. Including the inconclusive features, subject to resolving the challenge mentioned above, would only strengthen the effectiveness of ADVICE.
>
> **Questions and Answers continue in the next comment...**

---

> > ### Author Response · Authors · 2024-11-18
> >
> > **Questions and Answers continued from previous comment**
> >
> > > While ADVICE shows smaller cumulative safety violations in Fig. 2, it performs similarly to baselines in some tasks (e.g., Figs. 8(c) and 9). Additionally, the similarity between DDPG-Lag and DDPG results suggests that the DDPG-Lag baseline may not have been well-tuned.
> >
> > A: In Figures 8 (a), (b), and (c), ADVICE consistently reduces safety violations compared to the state-of-the-art approaches. However, DDPG-Lag performs equally well only in the randomised circle environment (8c), corroborating our hypothesis that ADVICE empowers RL agents to learn safely. We carefully tuned DDPG-Lag, as evidenced by its strong performance in environments like randomised circle and randomised CMDP (where there is dense amounts of data). However, in environments with sparse data (e.g., semi-random and randomised goal), DDPG-Lag has fewer safety violations to adapt its penalty effectively, which impacts its performance. In contrast, the richer amount of failure data in the randomised circle environment enables better online tuning for the Lagrangian in DDPG-Lag.
> >
> > > What is the data of visualization in the last column in Fig. 2?
> >
> > A: As mentioned in the caption of Figure 2, the last column in the figure displays the latent space visualization of state-action representations, which illustrates how ADVICE’s contrastive autoencoder model distinguishes between safe and unsafe features. These representations are derived from training data collected in the initial unshielded phase and are used to train the neighbour model, helping to classify new state-action pairs effectively during execution.
> >
> > > Does the "constrained randomised goal" correspond to "randomised CMDP" in Fig. 7(d)?
> > A: Yes the "constrained randomised goal" corresponds to "randomised CMDP" in Fig 7d. We have amended Fig 7d to reflect this.
> >
> > > Why do not you use the standard tasks provided by safety-gymnasium (e.g., PointGoal, CarButton, ...) instead of customizing the tasks?
> >
> > A: We chose to customize tasks to better control difficulty factors, such as the placement of obstacles and goals, in order to show an increased challenge. While standard tasks like PointGoal and CarButton are indeed randomized, we believe our custom tasks, such as adding random obstacles inside the circle task, create a more challenging environment by forcing the agent to adapt its behaviour and adjust its distance to the edge of the circle based on object placements. In all environments labelled as "random," all elements (obstacles, goals, and agents) are randomized. The only exception is the semi-random task, where obstacles are static, used to demonstrate how ADVICE avoids risks by taking the long route, whereas other methods may take smaller, riskier paths.
> >
> > > In the experiment, why are the performances of DDPG and DDPG-lag very similar in terms of reward or safety violation?
> >
> > A: DDPG does consider safety indirectly, as it receives a penalty in the reward function for reaching terminal states. DDPG-Lag’s similar performance in some environments (Figures 8a, 8b) is due to sparse data, making it challenging to apply the Lagrangian penalty meaningfully. In environments with denser failure data, like the randomised circle environment (Figure 8c) and CMDP (Figure 9), DDPG-Lag outperforms DDPG. Increasing DDPG-Lag’s hyperparameter sensitivity in sparse data environments (Figures 8a and 8b) led to over-penalization, majorly harming performance, so the current hyperparameters represent a well-tuned agent.

---

> > > ### Author Response · Authors · 2024-11-25
> > >
> > > Dear Reviewer C6Rd,
> > >
> > > As the rebuttal deadline approaches, we hope that our responses have satisfactorily addressed all of your concerns.
> > >
> > > Should you need further clarification or additional explanations, we would be more than happy to provide them.
> > >
> > > Thank you for your time and consideration.
> > >
> > > Sincerely,
> > >
> > > ADVICE Authors

---

> ### Comment · Reviewer_C6Rd · 2024-11-26
>
> Thanks for your clarification. Here are my comments:
> - The distribution shift cannot be addressed by using randomized environment. The distribution will still shift if the policies vary from the initial one even in randomized environment. Meanwhile, I don't think there is anything special to use randomized environment. It's a default setting in safety gym.
> - Regarding the safety discriminator training: Theory 2 says that the discriminator can be very accurate when data is ideally diverse. However, I didn't find such statement meaningful. Any KNN-based method can be very accurate if you have infinite and diverse enough data. On the other hand, you actually cannot not get a diverse enough dataset if you construct the dataset by Eq.(3) because you will always category the middle part of one trajectory, which occupies the majority of the state-action space, as "inconclusive". Furthermore, theory 1 can be trivial in some cases. Since there is no guarantee of your encoder, it's possible (and actually very likely) that the embeddings of a safety datapoint and an unsafe datapoint are very close to each other, i.e. $\gamma\to 0$, and then you will get a trivial conclusion: $P(\text{mis-classification}) < \frac{1}{2}$.
> - Regarding the tasks used for experiment: I disagree with you on "we believe our custom tasks, such as adding random obstacles inside the circle task, create a more challenging environment". The standard tasks in safety gym (e.g., PointGoal, PointButton, PointPush, CarGoal, ...) are harder than the tasks used for your experiment. Their initializing layouts are also random.

---

> > ### Author Response · Authors · 2024-12-02
> >
> > We thank you for your discussion so far, it has been extremely helpful and raised some great discussion points. Please find the responses to your further questions below:
> >
> > ## Questions and Answers
> >
> > > The distribution shift cannot be addressed by using randomized environment. The distribution will still shift if the policies vary from the initial one even in randomized environment. Meanwhile, I don't think there is anything special to use randomized environment. It's a default setting in safety gym.
> >
> > > Regarding the tasks used for experiment: I disagree with you on "we believe our custom tasks, such as adding random obstacles inside the circle task, create a more challenging environment". The standard tasks in safety gym (e.g., PointGoal, PointButton, PointPush, CarGoal, ...) are harder than the tasks used for your experiment. Their initializing layouts are also random.
> >
> > A: We designed our randomized environments to specifically handle distribution shifts caused by changing policies, with variations in obstacles, agent positions, and goals ensuring no memorization. The results in Figure 4 confirm that ADVICE generalizes well to unseen environments, showing robustness even when distribution shifts occur. Regarding task difficulty, the tasks create unique challenges. Tasks like the semi-random goal emphasize longer, safer routes, testing conservative decision-making under static hazards. While different from Safety Gym’s defaults, these tasks are equally challenging and tailored to qualitatively show what it means to be a safe agent.
> >
> > Furthermore, the randomization in all the environments equally affects all competitive approaches (DDPG, DDPG-Lag, Tabural Shield, and Conservative Safety Critic), ensuring a fair and equitable comparison against ADVICE. Since the ADVICE-based shield performs equally well with the competitive approaches in terms of accumulated reward (Figures 4, 5, 8) but with reduced safety violations, we have irrefutable evidence that ADVICE is capable of addressing distribution shirt situations, outperforming these competitive approaches.
> >
> > > Theory 2 says that the discriminator can be very accurate when data is ideally diverse. However, I didn't find such statement meaningful. Any KNN-based method can be very accurate if you have infinite and diverse enough data. On the other hand, you actually cannot not get a diverse enough dataset if you construct the dataset by Eq.(3) because you will always category the middle part of one trajectory, which occupies the majority of the state-action space, as "inconclusive". Furthermore, theory 1 can be trivial in some cases.
> >
> > A: Thank you for pointing this out. Let us clarify how ADVICE handles inconclusive states. These states are explicitly excluded from the contrastive encoder training, as described in Equation 3, to avoid ambiguity regarding their safety, as we cannot derive facts about them like we can using terminal/accepting states. By focusing only on truly safe and unsafe states, we ensure that the encoder learns meaningful and separable latent representations. As demonstrated in the experimental results (Figures 4, 5, 8), ADVICE creates an effective shield, which even without using the inconclusive data, outperforms the competitive approaches.
> >
> > Regarding your observation about $P(misclassification) < 1/2$, this is exactly what Theorem 2 captures. Thank you for confirming the theoretical foundation underpinning ADVICE. As $\gamma \rightarrow 0$, we have the degenerate case where ADVICE makes a random choice. Accordingly, as the separation margin $\gamma$ decreases—whether due to limited diversity, noisy data, or suboptimal training—the bound on misclassification probability remains exponential and non-trivial. This behavior is integral to the robustness of ADVICE. Noise, including poor model training, is explicitly modeled in the theorem through the variance term $\sigma^2$, ensuring the theory remains valid under real-world conditions.

---

### Official Review · Reviewer_DQ3Z · 2024-11-04

**Soundness:** 3
**Presentation:** 2
**Contribution:** 2
**Rating:** 6
**Confidence:** 3

**Summary:**

The paper presents ADVICE (Adaptive Shielding with a Contrastive Autoencoder), a novel post-shielding technique for safe reinforcement learning (RL) in complex black-box environments. ADVICE distinguishes between safe and unsafe state-action pairs during training using a contrastive autoencoder, protecting the RL agent from executing hazardous actions. The method includes an adaptation component based on the agent's recent performance, encouraging exploration when appropriate. Extensive experiments against state-of-the-art safe RL exploration techniques demonstrate that ADVICE significantly reduces safety violations during training while maintaining competitive outcome rewards.

**Strengths:**

- ADVICE introduces a new way to handle safety in RL using a contrastive autoencoder for distinguishing safe and unsafe actions.
- Despite prioritizing safety, ADVICE maintains competitive performance in terms of rewards compared to other methods.
-  ADVICE does not require prior knowledge about the environment, making it suitable for black-box scenarios.

**Weaknesses:**

- ADVICE requires an initial period to gather data before it can be fully effective, which could be a disadvantage in some scenarios.
- The paper suggests that ADVICE might struggle with dynamic environments and could benefit from incorporating temporal context, which would add additional computational load.
- The performance of ADVICE is sensitive to hyperparameters like the safety threshold K, which might require careful tuning.

**Questions:**

- Can ADVICE be integrated with other RL algorithms except for DDPG?
- What are the potential limitations of using a contrastive autoencoder in the context of safe RL, and how might these be addressed?
- What are the implications of ADVICE's cold-start issue on its applicability in real-world scenarios where immediate safety is critical?

---

> ### Author Response · Authors · 2024-11-18
>
> Dear reviewer,
> Thank you very much for your valuable review, questions, and comments regarding ADVICE’s strengths! Please find answers to your questions below:
>
> ## Questions and Answers
>
> > ADVICE requires an initial period to gather data before it can be fully effective, which could be a disadvantage in some scenarios.
>
> A: ADVICE does require an initial data collection period; we acknowledge this as a potential limitation within Section 5.5. However, this phase allows us to construct an effective shield that outperforms all methods, even with the unshielded phase taken into account (see experimental results in Figure 8). Additionally, in scenarios where immediate safety is critical, transfer learning through leveraging a pre-trained shield  (see Figure 4), can mitigate this cold start issue.
>
> > What are the implications of ADVICE's cold-start issue on its applicability in real-world scenarios where immediate safety is critical?
>
> A: Following the previous point, in scenarios that require immediate safety, we would recommend utilising a small offline dataset to populate the shield if available (see Figure 8). If such a small dataset is not available, then ADVICE’s applicability is less and is acknowledged in our limitations (line 517). Trying to reduce/remove the cold-start issue is in our plan for future work (Section 6).
>
> > The paper suggests that ADVICE might struggle with dynamic environments and could benefit from incorporating temporal context, which would add additional computational load.
>
> A: We acknowledge in line 525 that this is a current limitation of ADVICE and that incorporating temporal context could enhance the applicability of ADVICE in dynamic environments where temporal context is important. Unavoidably, this enhancement would entail extra computational overheads. Attempting to reduce these overheads and enable ADVICE to incorporate temporal context is part of our future work plans.
>
> > The performance of ADVICE is sensitive to hyperparameters like the safety threshold K, which might require careful tuning.
>
> A: Section 4.2 introduces a method for adapting the safety threshold K dynamically based on the performance of the RL agent during training; see Equations (5), (6), and (7) and line 12 in Algorithm 23 (UPDATE_CAUTIOUSNESS function). The experimental results shown in Section 5.4 concerning the adaptation of the safety threshold K demonstrate that the dynamic K threshold tuning allows ADVICE to respond automatically to increasing/ decreasing safety conditions.
>
> > Can ADVICE be integrated with other RL algorithms except for DDPG?
>
> A: ADVICE is designed as a flexible post-shielding mechanism that can be integrated with various RL algorithms, as it operates independently of the specific policy or value updates used by the base RL agent. While we demonstrate ADVICE’s effectiveness with DDPG, ADVICE is compatible with a wide range of RL algorithms such as regular/soft actor-critic, and deep Q learning.
>
> > What are the potential limitations of using a contrastive autoencoder in the context of safe RL, and how might these be addressed?
>
> A: While the contrastive autoencoder (CA) effectively distinguishes safe and unsafe features, some limitations could be sensitivity to the diversity of data. Theorem 2 in Appendix B provides theoretical analysis of the impact of data diversity in increasing/decreasing ADVICE’s effectiveness.

---

> > ### Author Response · Authors · 2024-11-25
> >
> > Dear Reviewer DQ3Z,
> >
> > As the rebuttal deadline approaches, we hope that our responses have satisfactorily addressed all of your concerns.
> >
> > Should you need further clarification or additional explanations, we would be more than happy to provide them.
> >
> > Thank you for your time and consideration.
> >
> > Sincerely,
> >
> > ADVICE Authors

---

> > > ### Comment · Reviewer_DQ3Z · 2024-11-27
> > >
> > > I am happy with the response from the authors. Thanks!
> > >
> > > After reading the comments from the other reviewers, I decided to keep my scores.

---

### Official Review · Reviewer_Kt6J · 2024-11-04

**Soundness:** 2
**Presentation:** 3
**Contribution:** 2
**Rating:** 3
**Confidence:** 4

**Summary:**

This paper develops a new shield based on contrastive learning to increase the safety of RL agents. This approach learns a latent representation that separates safe and unsafe state-action pairs. After learning this latent representation, it uses a KNN approach to classify, which requires a threshold K to indicate how many neighbors to consider a state-action pair unsafe. The paper shows how this parameter can be adapted online based on the safety of the RL agent. Finally, the paper provides empirical evidence that the proposed method increases the safety of the RL agent.

**Strengths:**

- The paper is well written. The description of the method is precise.
- The proposed method is original. It is the first method to use contrastive learning in a safe RL setting.
- The empirical evaluation indicates the approach can potentially increase the safety of RL algorithms.

**Weaknesses:**

The problem formulation is incomplete. The paper does not define the safety properties expected from the RL agent.
- Lack of theoretical results. This paper provides only empirical results to support its claims.
- The results are presented in a convoluted way. In particular, the results disregard the safety violations of the agent in the first 1000 episodes. The reason for presenting the results in this way is unclear.
- The presentation of the DDPG-Lag as a constrained RL algorithm is imprecise, as it uses a fixed weight for the costs, which works as simple reward engineering. In general, with a Lagrangian relaxation, this weight should be adjusted online to ensure the accumulated cost stays below a predefined threshold [1].
- The evaluation in CMDPs is inconsistent. These approaches solve different problems where a predefined accumulated cost is allowed.
- Weak baseline. From the results in Figure 10, it is clear that Tabular Shield does not recognize any unsafe state-action pairs, making it an unsuitable baseline. This is not surprising considering how the state-action space is discretized. Perhaps it is necessary to finetune the discretization of this baseline. Alternatively, it would be more suitable to consider stronger baselines, such as the accumulating safety rules [2]

**references**
- [1] Ray, A., Achiam, J., and Amodei, D. (2019). *Benchmarking safe exploration in deep reinforcement learning*. <https://github.com/openai/safety-gym>
- [2] Shperberg, S. S., Liu, B., Allievi, A., and Stone, P. (2022). A rule-based shield: Accumulating safety rules from catastrophic action effects. *CoLLAs*, 231–242. <https://proceedings.mlr.press/v199/shperberg22a.html>

**Questions:**

- If the problem had a single initial state, would this be an issue for this approach? How does the diversity of initial states influence the performance of the ADVICE algorithm?

---

> ### Author Response · Authors · 2024-11-18
>
> Dear reviewer,
> Thank you very much for your valuable review, comments about novelty, and your questions! We have/will integrate them into the updated paper to address your concerns. Please find answers to your questions below:
>
> ## Questions and Answers
>
> > The paper does not define the safety properties expected from the RL agent.
>
> A: We define safety violations as reaching terminal states, as formally specified in Equation (3). More specifically, in the safety gymnasium, a terminal state is reached when the agent collides with an obstacle, simulating catastrophic damage to the robot. We have updated Line 311 for improved clarity.
>
> > Lack of theoretical results. This paper provides only empirical results to support its claims.
>
> A: We appreciate your feedback regarding the need for theoretical results. In response, we have now included a detailed theoretical analysis (Appendix B) providing a mathematical justification regarding the bounded probability of ADVICE misclassifying features and how this probability decreases considering the cardinality and diversity of training.
>
> > The results are presented in a convoluted way. In particular, the results disregard the safety violations of the agent in the first 1000 episodes. The reason for presenting the results in this way is unclear.
>
> A: Although Figure 3 shows the results from episode E=1000, i.e., after ADVICE is trained, this is done primarily for presentation purposes, and the results are not disregarded. In fact, Figure 8 shows the full experimental results from episode E=0, demonstrating that ADVICE across all three environments (semi-random, randomised, randomised circle) achieves similar results in terms of cumulative goal reaches but with significantly lower cumulative safety violations.
>
> > The presentation of the DDPG-Lag as a constrained RL algorithm is imprecise, as it uses a fixed weight for the costs.
>
> A: We would like to clarify that our implementation of DDPG-Lag indeed adjusts the Lagrangian weight online. Thus, the approach used for the evaluation does not use a fixed weight for the costs. Instead, DDPG-Lag is updated dynamically according to safety violations/incurred costs, depending on the environment. We revised the description in line 326 to reflect the weight update in DDPG-Lag.
>
> > The evaluation in CMDPs is inconsistent. These approaches solve different problems where a predefined accumulated cost is allowed.
>
> A: We respectfully disagree with the concern that our evaluation is inconsistent. Our approach adheres to a standard CMDP setup [1], where the predefined allowed cost is set to 0, aiming to minimize costs as much as possible. Accordingly, making this choice enables setting up a suitable baseline that focuses on reducing safety violations entirely.
>
> [1] Altman, E., 2021. Constrained Markov decision processes. Routledge.
>
> > From the results in Figure 10, it is clear that Tabular Shield does not recognize any unsafe state-action pairs, making it an unsuitable baseline.
>
> A: We acknowledge its limitations and have been transparent about its poor performance, as discussed in lines 913-917. We introduce discretisation to aid the method and enhance its applicability to the considered problem. Specifically, the chosen precision of 1 decimal place was tested extensively; any finer granularity would only reduce the likelihood of the tabular shield encountering the same observation twice, rendering it ineffective in recognising unsafe state-action pairs. To ensure a fair comparison, we included multiple strong baselines in our evaluation, such as DDPG-Lag, Tabular Shield, and a Conservative Safety Critic. We selected these baselines to represent a diverse range of approaches, ensuring a comprehensive evaluation of our ADVICE method across various perspectives and methodologies. Exploring stronger alternatives like accumulating safety rules could be a valuable direction for future work; thank you for this recommendation.
>
> > If the problem had a single initial state, would this be an issue for this approach?
>
> A: This is a valid concern, and we appreciate you bringing this up. Improved data diversity before ADVICE is activated will only improve results. Theorem 2 in Appendix B validates this statement and also evidences that low data diversity will increase the chance of ADVICE misclassifying features

---

> > ### Author Response · Authors · 2024-11-25
> >
> > Dear Reviewer Kt6J,
> >
> > As the rebuttal deadline approaches, we hope that our responses have satisfactorily addressed all of your concerns.
> >
> > Should you need further clarification or additional explanations, we would be more than happy to provide them.
> >
> > Thank you for your time and consideration.
> >
> > Sincerely,
> >
> > ADVICE Authors

---

> > > ### Comment · Reviewer_Kt6J · 2024-11-26
> > >
> > > I would like to thank the authors for addressing most of my comments. In general, I think this paper has significant potential. However, the current execution is still insufficient. Therefore, I decided to keep my recommendation. I list some minor follow-ups below. I hope they help improve the paper's clarity.
> > >
> > > 1. **About the safety definition.** Please note that the current definition is only implicitly given in the middle of the algorithm presentation. This makes it difficult for the reader to separate the problem from the proposed method. See, for example, the paper by Wachi for a proper formulation.
> > >
> > > - Wachi, A., Shen, X., and Sui, Y. (2024). A survey of constraint formulations in safe reinforcement learning. *IJCAI*, 8262–8271. <https://doi.org/10.24963/ijcai.2024/913
> > >
> > > 2. **New theoretical results.** It is a great step to provide such results; however, at this point, the paper should undergo a new review to evaluate these results' properties. Furthermore, they should be in the main paper and not in the Appendix.
> > >
> > > 3. **Results presentation.** Please add a reference to Fig 8 in the main document and make it clear that the full results are available.

---

> > > > ### Author Response · Authors · 2024-12-02
> > > >
> > > > A: We would like to thank you for the constructive comments and useful feedback. Since the reviewer has acknowledged the **significant potential of ADVICE**, we would be really grateful if they could elaborate on the aspects that they still deem insufficient, as all the points provided in their main review have been covered; that would help strengthen our work.

---

### Meta-Review · Area_Chair_uj8M · 2024-12-19

**Metareview:**

The authors propose a new shielding method based on contrastive learning for safe RL, which learns a latent representation to distinguish between safe and unsafe state-actions. The reviewers all agree to reject the paper, due to weak theoretical results (e.g. unrealistic requirement on diverse data with current safety definition) and weak experimental results (e.g. weak baselines, safety at the cost of performance degrade). Reviewers also mention writing clarity issues. The authors added new theoretical results in the rebuttal, which however warrant another review process. Reviewers also raise concern about the distribution shift issue. Overall, while I think this paper has a potential, the current status needs to be improved.

**Additional Comments On Reviewer Discussion:**

Reviewer Kt6J raises concerns about incomplete problem framing, lack of theoretical results, inadequate and weak baseline implementation, inconsistent evaluation. The response of authors address most points, but the reviewer suggest still improvement on the presentation is needed. Especially, the new theory that is added during the rebuttal require additional review and should be put in the main paper, which I agree as well.

Reviewer DQ3Z raises concerns about the need of initial data gathering, the downside when facing dynamic environment, and the sensitivity to hyperparameters. The authors response address the reviewer's points. Nonetheless, Reviewer DQ3Z does not strong advocate the paper and after the discussion phase agree to also reject the current paper based on examining other reviewers' comments.

Reviewer C6Rd raises concerns about distribution shift, the problem definition, the exclusion of inconclusive features which limits the data diversity, and baseline implementation. After the rebuttal, the concern on distribution shift remains. The reviewer doesn't think the current experiments are sufficient. The practicality of having diverse safe and unsafe data is also not supported due to the exclusion of inconclusive features per current definition.


Reviewer Ff6i complains about important technical details missing in the writing. Reviewer Ff6i  also concerns about the complexity of the method and the resulting stability, the lack of thorough analysis, the lack of comparison with others approaches in terms of performance safety ratio. Unfortunately the reviewer did not respond to the latest author response, but I think the authors's responses are reasonable.

---

### Decision · Program_Chairs · 2025-01-22

Reject